# Strand-specific single-cell methylomics reveals distinct modes of DNA demethylation dynamics during early mammalian development

Maya Sen[1,8], Dylan Mooijman[1,7,8], Alex Chialastri [2,3,8], Jean-Charles Boisset[1], Mina Popovic[4], Björn Heindryckx [4], Susana M. Chuva de Sousa Lopes [4,5], Siddharth S. Dey [2,3,6 ✉] & Alexander van Oudenaarden [1✉]

DNA methylation (5mC) is central to cellular identity. The global erasure of 5mC from the parental genomes during preimplantation mammalian development is critical to reset the methylome of gametes to the cells in the blastocyst. While active and passive modes of demethylation have both been suggested to play a role in this process, the relative contribution of these two mechanisms to 5mC erasure remains unclear. Here, we report a single-cell method (scMspJI-seq) that enables strand-specific quantification of 5mC, allowing us to systematically probe the dynamics of global demethylation. When applied to mouse embryonic stem cells, we identified substantial cell-to-cell strand-specific 5mC heterogeneity, with a small group of cells displaying asymmetric levels of 5mCpG between the two DNA strands of a chromosome suggesting loss of maintenance methylation. Next, in preimplantation mouse embryos, we discovered that methylation maintenance is active till the 16-cell stage followed by passive demethylation in a fraction of cells within the early blastocyst at the 32-cell stage of development. Finally, human preimplantation embryos qualitatively show temporally delayed yet similar demethylation dynamics as mouse embryos. Collectively, these results demonstrate that scMspJI-seq is a sensitive and cost-effective method to map the strand-specific genome-wide patterns of 5mC in single cells.

[1] Oncode Institute, Hubrecht Institute-KNAW (Royal Netherlands Academy of Arts and Sciences) and University Medical Center Utrecht, Utrecht, The Netherlands. [2] Department of Chemical Engineering, University of California Santa Barbara, Santa Barbara, CA 93106, USA. [3] Center for Bioengineering, University of California Santa Barbara, Santa Barbara, CA 93106, USA. [4] Ghent-Fertility and Stem cell Team (G-FaST), Department of Reproductive Medicine, Ghent University Hospital, 9000 Ghent, Belgium. [5] Department of Anatomy and Embryology, Leiden University Medical Center, 2333 ZC Leiden, The Netherlands. [6] Neuroscience Research Institute, University of California Santa Barbara, Santa Barbara, CA 93106, USA. [7] Present address: Developmental Biology Unit, European Molecular Biology Laboratory, Heidelberg, Germany. [8] These authors contributed equally: Maya Sen, Dylan Mooijman, Alex Chialastri. ✉email: sdey@ucsb.edu; a.vanoudenaarden@hubrecht.eu

In mammalian systems, DNA methylation (5-methylcytosine or 5mC) is a key epigenetic modification that is typically stably inherited from mother to daughter cells[1]. This property of 5mC plays an important role in facilitating the propagation of cellular identity through cell divisions and restricting the developmental potential of terminally differentiated cells[1,2]. Consequently, during preimplantation mammalian development, DNA methylation patterns on the terminally differentiated paternal sperm and maternal egg genomes are erased post-fertilization at a genome-wide scale to revert cellular memory towards an undifferentiated state in the blastocyst[3]. Therefore, understanding the mechanisms underlying global DNA demethylation dynamics is central to understanding the emergence of pluripotent cells during early development.

Removal of 5mC can proceed through two alternate mechanisms —passive and active demethylation. Methylated cytosines, within a CpG dinucleotide context are typically copied over to the newly synthesized DNA strands during genome replication by the maintenance methyltransferase, DNMT1[4]. Passive demethylation relies on loss of 5mC through replicative dilution, in which inhibition of DNA methylation maintenance results in a reduction of 5mC levels after cell division and can be detected through asymmetric levels of 5mC on the two DNA strands of a chromosome. Alternatively, active mechanisms of 5mC erasure occur via conversion of 5mC to 5-hydroxymethylcytosine (5hmC) and other oxidized derivatives, which are not recognized by the DNA maintenance methylation machinery and are subsequently removed by base-excision repair pathways[5–7]. While early immunofluorescence-based studies revealed that the paternal genome undergoes active demethylation through conversion to 5hmC in the zygote, the maternal genome was presumed to undergo passive demethylation through the lack of DNMT1 activity during replication[8–11]. Advances in biochemistry, next-generation sequencing, and mass-spectroscopy-based studies improved upon this coarse quantification of methylation dynamics to show that the orthogonal regulation of demethylation by active and passive mechanisms for the two parental genomes was not as distinct as suggested by these early studies. For example, it was later shown that while DNMT1 is mostly cytoplasmic during these early stages of development, low levels of a *Dnmt1* isoform, DNMT1s, together with UHRF1 is observed in the nucleus, raising the possibility that 5mC is maintained on the maternal genome[12–19]. However, the conclusions in these recent studies were partly based on bulk bisulfite-sequencing-based methods that could not directly distinguish between active vs. passive demethylation, and therefore the relative contribution of these two mechanisms to 5mC reprogramming remains poorly understood.

## Results

**Strand-specific quantification of 5mC using scMspJI-seq.** To distinguish between active and passive mechanisms of demethylation requires strand-specific detection of 5mC in single cells. While asymmetric levels of 5mC between two DNA strands of a chromosome would indicate passive demethylation, the global loss of methylation coupled with symmetric levels of 5mC between two DNA strands would indirectly imply active demethylation (Fig. 1a)[20]. Therefore, to identify the mechanisms regulating DNA demethylation dynamics, we developed a method called scMspJI-seq to strand-specifically quantify 5mC on a genome-wide scale in single cells. Single cells are isolated into 384-well plates by fluorescence activated cell sorting or manual pipetting. All downstream steps are subsequently performed using a liquid-handling platform (Nanodrop II, BioNex Solutions). Following cell lysis and protease treatment to remove chromatin, 5hmC sites in genomic DNA (gDNA) are glucosylated using T4 phage β-glucosyltransferase (T4-βGT) (Fig. 1b). This modification blocks downstream detection of 5hmC and

therefore, enables detection of only 5mC in scMspJI-seq. Next, the restriction enzyme MspJI is added to the reaction mixture that recognizes ^mCNNR sites in the genome and creates double-stranded DNA breaks 16 bp downstream of the methylated cytosines leaving a 4-nucleotide 5′ overhang[21]. Thereafter, double-stranded DNA adapters containing a 4-nucleotide 5′ overhang are ligated to the fragmented gDNA molecules. These double-stranded DNA adapters, similar in design to those previously developed by us, contain a cell-specific barcode, a random 3 bp unique molecule identifier (UMI) to label individual 5mC sites on different alleles, a 5′ Illumina adapter and a T7 promoter[22,23]. The ligated molecules are then amplified by in vitro transcription and used to prepare Illumina libraries as described previously, enabling the processing of hundreds to thousands of single cells per day (Fig. 1b)[22,23].

To validate the method, we first applied scMspJI-seq to single E14TG2a (E14) mouse embryonic stem cells (mES) cells. As reported previously, we found that MspJI cuts gDNA 16 bp downstream of the methylated cytosine (Supplementary Fig. 1)[21]. We detected between 212,000 and 977,000 unique 5mC sites per cell, with a median of 484,000 5mC sites per cell (Supplementary Fig. 2). Further, we found that 97.2% of the 5mCpG sites detected by scMspJI-seq in single cells overlapped with methylated sites observed in bulk bisulfite sequencing of E14 gDNA (Supplementary Fig. 3a). Similarly, we found that averaged single-cell data from scMspJI-seq correlates well with the bulk bisulfite methylome (Pearson $r = 0.84$) (Supplementary Fig. 3b)[24]. Furthermore, while we observed that the genome-wide distribution of 5mC over different genomic elements in scMspJI-seq was similar to that observed in bisulfite sequencing, we also found that scMspJI-seq shows a slight preference for detection of 5mC sites within genomic regions that have a lower density of CpG sites (Supplementary Fig. 4,5). This possibly occurs as our method is dependent on the digestion of the genome around methylated cytosines, reducing the likelihood of detecting closely spaced 5mC sites. However, both scMspJI-seq and bisulfite sequencing captured similar genome-wide landscapes of 5mC at a variety of genomic elements. For example, we observed similar gene body methylome profiles as well as the expected hypomethylation of CpG islands (CGI) and transcription start sites (TSS) using both methods (Supplementary Fig. 6). In addition, compared to single-cell bisulfite sequencing that detects a combination of 5mC and 5hmC sites, a distinct feature of scMspJI-seq is that it can identify only 5mC in the genome by blocking the detection of 5hmC sites using T4-βGT. By combining scMspJI-seq data with scAba-seq results, we were able to estimate the false-positive detection rate of 5hmC to be around 1.1% (Supplementary Fig. 7)[22]. Most importantly, due to the maintenance activity of DNMT1 in E14 cells, we observed similar levels of 5mC on both DNA strands of a chromosome in single cells, as expected (Supplementary Fig. 8a). To quantify the strand-specific distribution of 5mCpG on each chromosome of a single cell, we defined a metric called as strand bias (denoted by $f$), which is the ratio of the number of 5mCpG sites detected on the plus strand divided by the total number of 5mCpG sites detected on both the plus and minus strands. Finally, to ensure that scMspJI-seq can detect differences in 5mCpG distribution between the two strands, and to confirm that the observed strand bias of 0.5 in E14 cells results from the maintenance activity of DNMT1, we used CRISPR-Cas9 to knockout *Dnmt1*. We observed a dramatic increase in strand bias in E14 cells without *Dnmt1*, strongly suggesting that our technology provides a sensitive readout of strand-specific methylation and the ability to distinguish between passive and active demethylation (Supplementary Fig. 8b).

**mES cells display heterogeneity in strand-specific 5mC.** During preimplantation development, the maternal and paternal

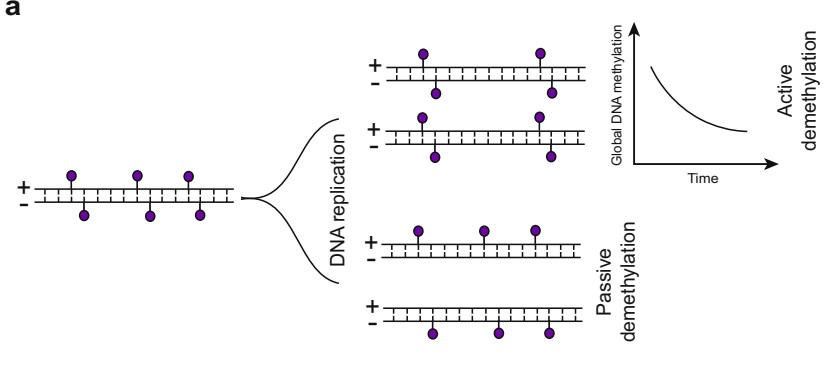

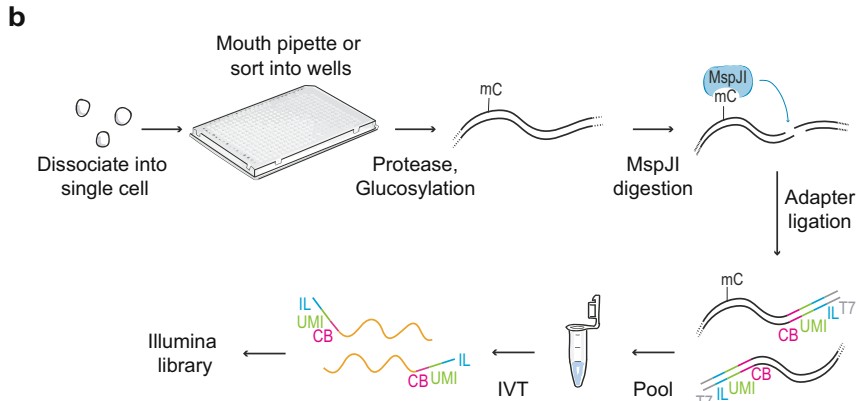

**Fig. 1 Schematic of scMspJI-seq. a** DNA methylation maintenance can be probed using strand-specific quantification of 5mC in single cells. Cells displaying symmetric levels of 5mCpG on both DNA strands of a chromosome coupled with a global temporal loss of 5mCpG indicates active demethylation, whereas loss of methylation maintenance with asymmetric levels of 5mCpG between the two DNA strands indicates passive demethylation. **b** Single cells isolated by FACS or manual pipetting are deposited into 384-well plates and lysed. Following protease treatment to strip off chromatin and blocking of 5hmC sites by glucosylation, MspJI is used to recognize 5mC sites and cut gDNA 16 bp downstream of the methylated cytosine. After ligating double-stranded adapters—containing a cell-specific barcode (CB, pink), a random 3 bp unique molecule identifier to label individual 5mC sites on different alleles (UMI, green), 5′ Illumina adapter (IL, blue) and T7 promoter (T7, gray)—to the fragmented gDNA, molecules from all single cells are pooled and amplified by in vitro transcription. The amplified RNA molecules are used to prepare scMspJI-seq libraries and sequenced on an Illumina platform.

genomes display dramatically different 5mC erasure dynamics, and therefore we next wanted to test our ability to quantify strand-specific 5mC at the resolution of individual alleles. As the single-cell measurements in E14 cells did not provide allele-specific detection of 5mC for each chromosome, we applied scMspJI-seq to hybrid serum grown mES cells (CAST/EiJ × 129/Sv background)[23]. While the majority of cells displayed methylation maintenance as expected, we surprisingly observed a small population of cells that showed strong 5mC strand bias (Fig. 2). For example, cell 562 displayed similar levels of 5mCpG on the two DNA strands of chromosomes across both alleles (Fig. 2a), whereas cell 216 showed substantially different levels of 5mC on each DNA strand of a chromosome (Fig. 2b). Pearson correlation coefficients (*r*) between the plus and minus strands of individual cells show that while a majority of cells displayed high correlation, a small subset of cells were weakly correlated, suggesting loss of methylation maintenance in these cells (Fig. 2c). Allele-specific 5mCpG strand bias further revealed the existence of two epigenetically distinct population of mES cells (Fig. 2d). Taken together with the E14 cells, these results highlight that in the absence of allele-specific measurements, strand-specific 5mC quantification is averaged across both alleles, potentially obscuring a detailed view of the methylation status of the genome.

Finally, we find that these two distinct 5mC strand bias patterns are also observed at a sub-chromosomal resolution, suggesting that this is a genome-wide phenomenon that potentially arises from differential methylation maintenance between individual mES cells (Fig. 2e).

To validate this cell-to-cell heterogeneity in 5mC strand bias, we reanalyzed data from a recent study that quantified 5mC in single cells using bisulfite sequencing, a method that can potentially also be used to infer strand-specific 5mC[25,26]. In agreement with our findings using scMspJI-seq, reanalysis of the published dataset also revealed hybrid mES cells with similar levels of 5mC on the plus and minus strands, and a small fraction of cells with substantially different levels of 5mC on the two strands of a chromosome (Fig. 3). These results validate our previous observation of two distinct mES cell populations with and without 5mC strand bias (Fig. 2).

**Embryos display distinct modes of demethylation dynamics.**
After establishing this method, we next used scMspJI-seq to gain a deeper understanding of the 5mC erasure dynamics during preimplantation mouse development as the mechanistic details regulating this genome-wide reprogramming remains unclear

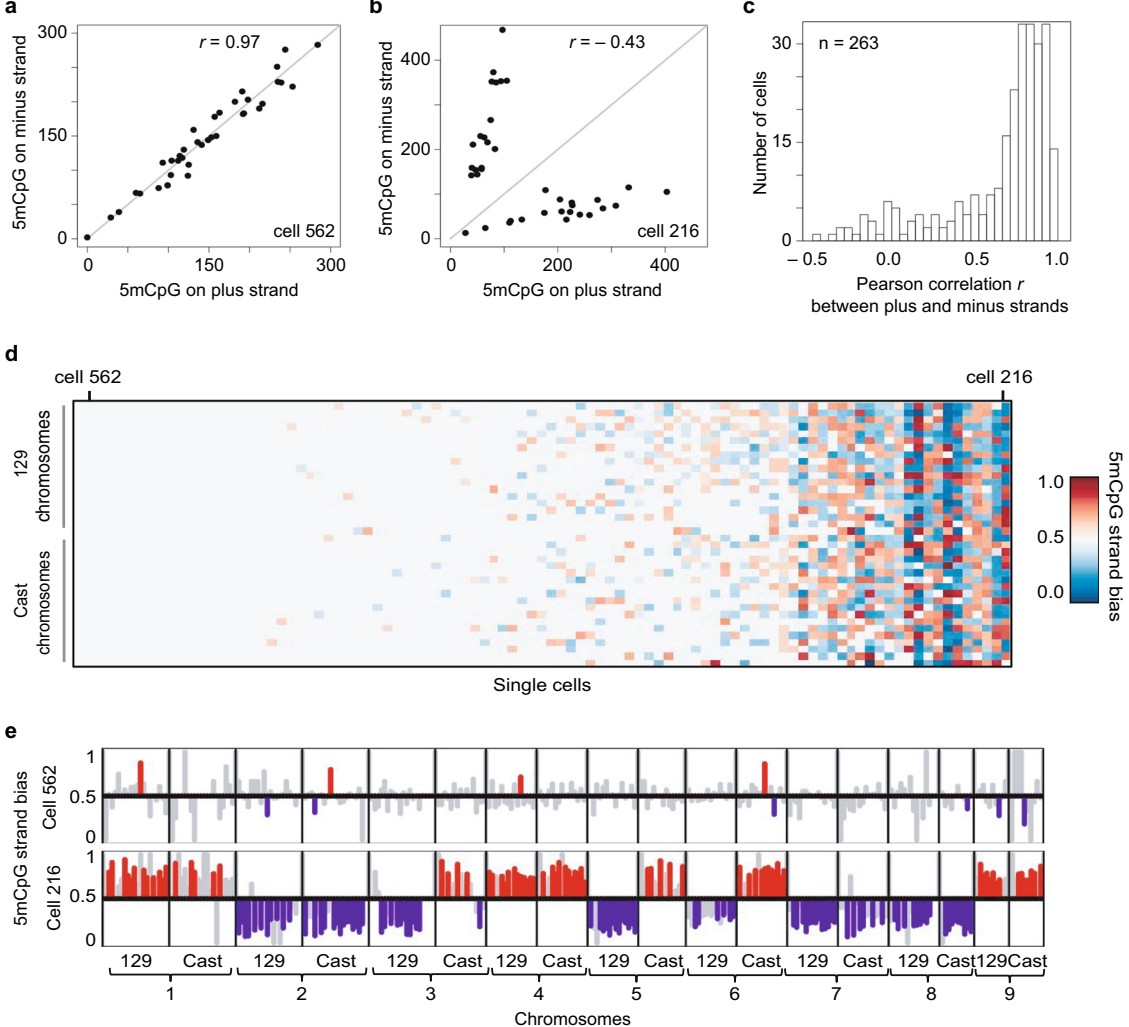

**Fig. 2 Cell-to-cell heterogeneity in genome-wide strand-specific methylome landscapes in mES cells. a** An example of a mES cell (cell #562) processed by scMspJI-seq shows similar amounts of 5mCpG on both the plus and minus strand of each chromosome. **b** Another mES cell (cell #216) with asymmetric amounts of 5mCpG between the plus and minus strands of each chromosome. **c** Histogram of Pearson correlations between the 5mCpG levels on the plus and minus stands over all chromosomes in a cell show that while a majority of cells have similar amounts of 5mCpG on both strands (high Pearson correlation), a small fraction of cells display unequal levels of 5mCpG between the two strands of each chromosome (low Pearson correlation). **d** Ordered heatmap showing 5mCpG strand bias per chromosome for the maternal and paternal alleles in individual mES cells. **e** 5mCpG strand bias of cell #526 (top) and cell #216 (bottom) for 10 Mb bins along the first 9 chromosomes are shown with statistically significant ($P < 0.05$, likelihood ratio test) strand biases towards the plus and minus strands shown in red and blue, respectively. Strand biases of bins that are not statistically significant are shown in gray ($P > 0.05$, likelihood ratio test).

from previous work. Early immunofluorescence-based studies showed that 5mC marks on the paternal genome are converted to 5hmC in the zygote[8–11]. As 5hmC is not maintained through cell division, and can be further oxidized to be removed by cytidine deaminase and base-excision repair pathways, the paternal genome is effectively demethylated from the 1-cell to early blastocyst stage (approximately E3.5 or 32-cell stage) of development[7]. These same studies also reported that the maternal genome retains 5mC in the zygote[8–11]. This observation together with reports that DNMT1 is primarily cytoplasmic during these early cell divisions, indirectly suggested that the maternal genome is passively demethylated through a lack of maintenance methylation[27–30]. However, later studies showed the existence of two isoforms of *Dnmt1*, with the lowly abundant DNMT1s isoform present in the nucleus of blastomeres[31–33]. Thus, it remains unclear the extent to which the maternal genome is passively demethylated during these early stages. Further, more recently, bulk 5mC and 5hmC sequencing during these early stages have

shown that the maternal genome also carries 5hmC marks, suggesting that the maternal genome also undergoes partial active demethylation[13]. As the mechanisms underlying this critical process of 5mC erasure during embryonic development remains unclear, we used strand-specific detection of 5mC in single cells to probe the dynamics of demethylation more closely.

We performed scMspJI-seq on hybrid mouse embryos (CAST/EiJ × C57BL/6 background) from the 2- to 32-cell stage of development. In contrast to previous studies that suggested passive demethylation of the maternal genome due to cytoplasmic localization of DNMT1, experiments in 2-cell hybrid mouse embryos surprisingly revealed that 5mCpG on the maternal genome shows a tight strand-bias distribution centered around 0.5, implying similar amounts of the mark of both DNA strands and that DNMT1-mediated methylation maintenance is active at this stage (Fig. 4a and Supplementary Fig. 9a). To ensure that this lack of strand bias in the maternal genome at the 2-cell stage is not a technical artifact or a consequence of high de novo

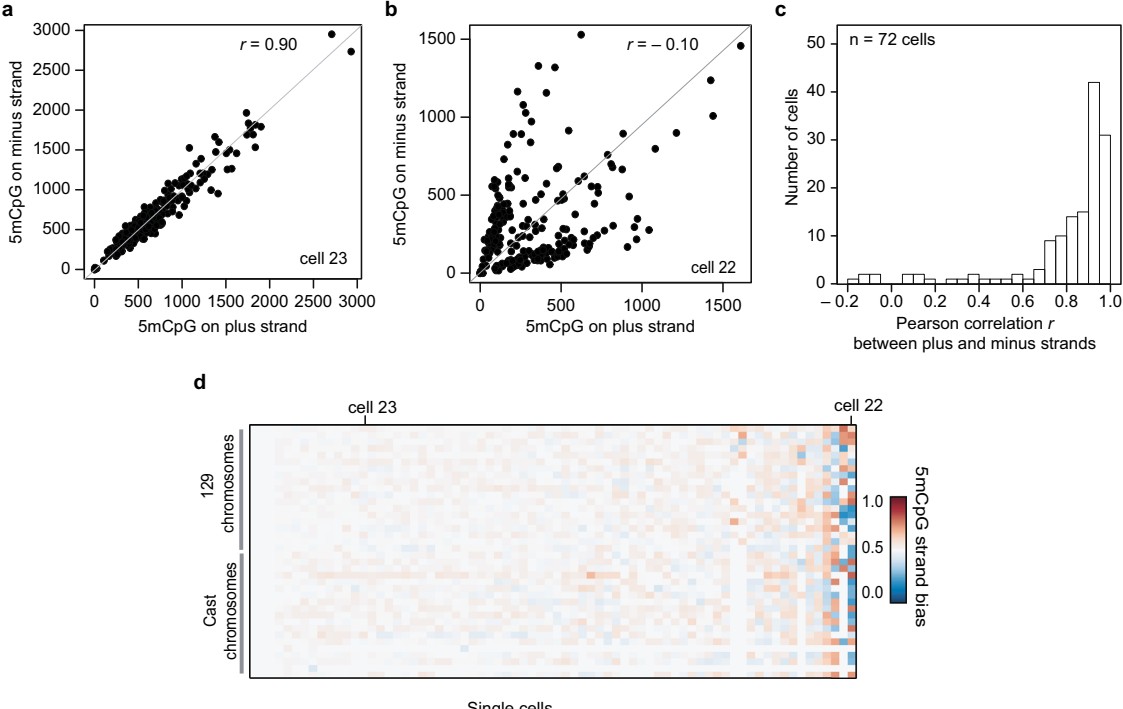

**Fig. 3 Variability in strand-specific 5mCpG profiles in mES cells. a** A representative mES cell (cell #23) with similar amounts of 5mCpG within 10 Mb bins on both DNA strands. **b** Another representative mES cell (cell #22) with unequal amounts of 5mCpG between the two DNA strands for 10 Mb bins. **c** Histogram of Pearson correlations between the 5mCpG levels on the plus and minus stands over the entire genome (10 Mb) in a cell. **d** Ordered heatmap showing 5mCpG strand bias per chromosome for maternal and paternal alleles in individual mES cells ($n = 72$). The results in this figure are based on strand-specific reanalysis of single-cell bisulfite sequencing data obtained from previous work by Clark et al.[25].

methylation activity of DNMT3a/3b, we quantified the levels of 5mCpA, the most abundant non-CpG methylation, in these cells. Non-CpG methylation is not a substrate for DNMT1 and is deposited on the genome as a result of the activity of the de novo methyltransferases, DNMT3a and DNMT3b[34–36]. In the 2-cell embryos, we found that 5mCpA on the maternal genome showed a bimodal pattern of strand-bias distribution, suggesting that the lack of strand bias observed for 5mCpG is possibly a result of the maintenance activity of DNMT1 and not a consequence of high de novo methylation rates by DNMT3a/3b (Fig. 4b and Supplementary Fig. 9b). Further, we have previously shown that bimodal strand-bias distributions for 5hmC in 2-cell mouse embryos arises from the slow kinetics of Tet activity and can be used to identify sister cells[22,37]. This is because 5hmC is not maintained through cell divisions and new DNA strands have lower levels of 5hmC than older strands, resulting in sister cells exhibiting anti-correlated strand bias patterns over all the chromosomes in a cell. Similarly, as 5mCpA is not maintained through cell division, we found that the strong anti-correlation in 5mCpA between chromosomes of single cells can be used to identify sister cells (Supplementary Fig. 9c, d). These results further imply that at the 2-cell stage of development the kinetics of de novo methylation by DNMT3a and DNMT3b is slow (Fig. 4b). Taken together, these experiments provide preliminary evidence that the similar levels of 5mCpG found on both DNA strands of chromosomes in 2-cell blastomeres is a result of DNMT1 maintenance activity.

Quantifying the dynamics of demethylation beyond the 2-cell stage, we observed for both the maternal and paternal genomes that a majority of chromosomes displayed no significant 5mCpG strand bias up to the 16-cell stage (Fig. 4a and Supplementary Fig. 9a). Surprisingly, beyond the 16-cell stage, we observed a widening of the 5mCpG strand-bias distribution, suggesting reduced DNMT1

maintenance activity (Fig. 4a and Supplementary Fig. 9a). These experiments suggest two distinct phases during preimplantation mouse development—an initial period of DNMT1-mediated maintenance methylation followed by passive demethylation. Finally, we observed that the 5mCpG strand-bias distribution at the 32-cell stage is trimodal. Performing $k$-means clustering on the 5mCpG strand bias in these single cells identified two distinct groups of cells as inferred by the mean silhouette scores—a population with no strand bias and another population with a bimodal strand-bias distribution (Fig. 4c, d). Further, within the bimodal population, we observed pairs of cells for which all chromosomes were strongly anti-correlated, suggesting that these pairs are sister cells (Fig. 4e and Supplementary Fig. 9e). These observations reveal the existence of significant cell-to-cell heterogeneity in the genome-wide methylome landscapes of cells within the early blastocyst. Taken together, these results suggest maintenance methylation is active till the 16-cell stage and that from the 16- to 32-cell stage, a fraction of cells within the embryo show strong 5mCpG strand bias and undergo passive demethylation.

Finally, to conclusively demonstrate that the absence of 5mCpG strand bias up to the 16-cell stage arises from DNMT1-mediated maintenance methylation, we performed bulk hairpin bisulfite sequencing on non-hybrid preimplantation mouse embryos. A hallmark of DNMT1-mediated methylation is that both cytosines in a CpG dyad are symmetrically methylated and therefore we performed bulk hairpin bisulfite sequencing that enables interrogation of the methylation status of CpG dyads[38]. We observed that the fraction of symmetrically methylated CpG dyads in the genome is high up to the 16-cell stage, with a dramatic reduction at the 32-cell stage (that is matched by an increase in hemi-methylated CpG dyads at this stage), thereby demonstrating that maintenance methylation is active initially and is followed by passive demethylation at the 32-cell stage (Fig. 4f).

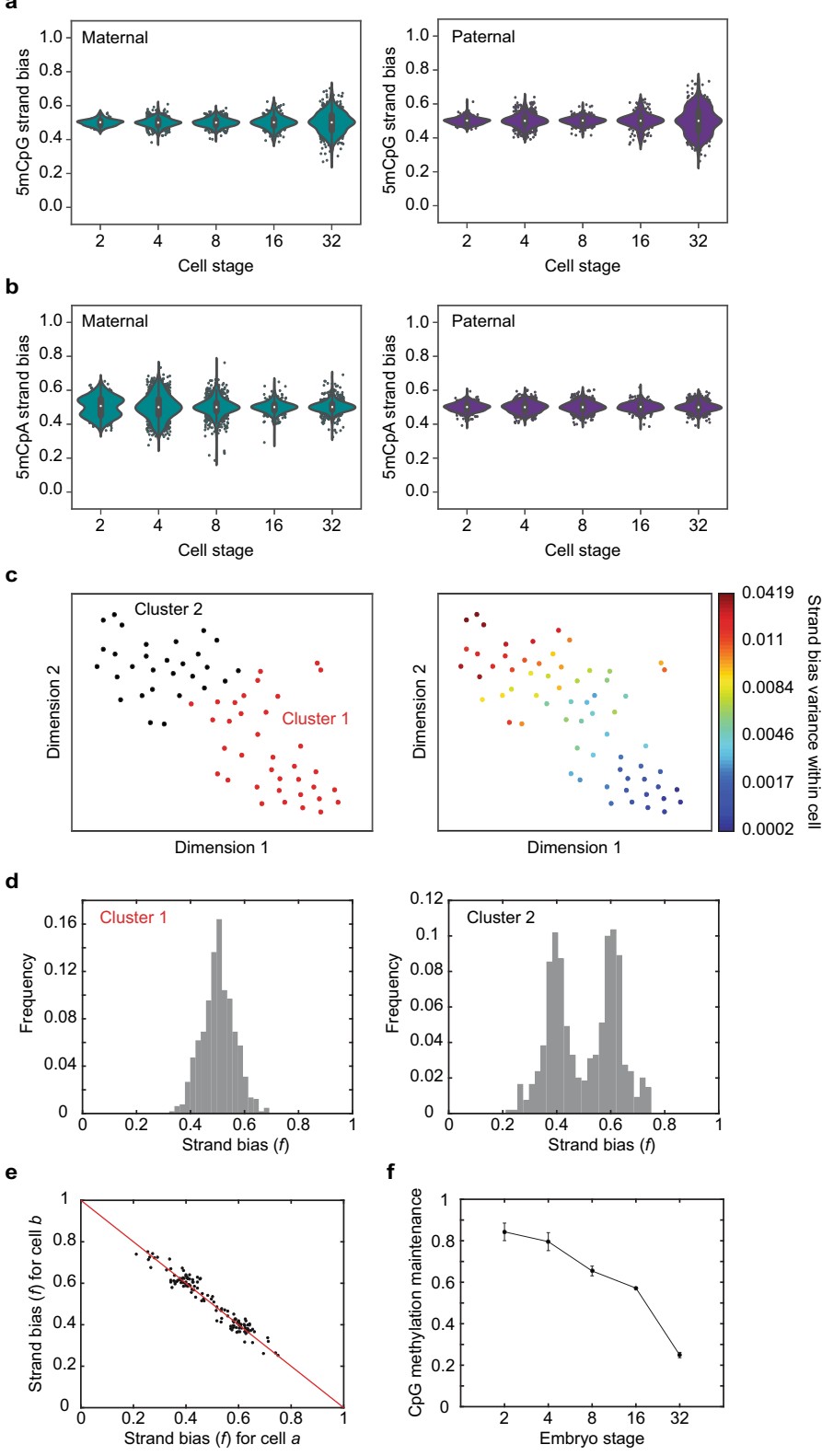

We finally extended scMspJI-seq to explore the dynamics of global demethylation in human preimplantation embryos, ranging from developmental day 2 to 7. Studies in human preimplantation embryos have shown temporally slower, yet similar developmental dynamics to mouse embryos[39]. Despite lacking allelic information, our results suggest that the mouse and human 5mCpG demethylation dynamics are similar, with an initial phase till the 16-cell stage displaying a tight 5mCpG strand-bias distribution centered around 0.5, followed by an increase in strand bias in a small fraction of cells from the 32- to 128-cell stage (Fig. 5a and Supplementary Fig. 10a). This is consistent with previous immunostainings in human preimplantation embryos that show a decrease in DNMT1 protein levels between day 5 and day 6 blastocysts[40,41]. Further, 5mCpA strand-

**Fig. 4 DNA demethylation dynamics in preimplantation mouse embryos. a** Violin plots of 5mCpG strand bias for both the maternal (left) and paternal (right) genome show a tight distribution centered around $f = 0.5$ till the 16-cell stage and a wider distribution at the 32-cell stage of development ($n = 332$ single cells from 42 embryos). **b** For the maternal genome (left), 5mCpA strand bias shows a bimodal distribution at the 2-cell stage that moves towards a tight unimodal distribution by the 32-cell stage of development. The paternal genome (right) shows a unimodal distribution centered at $f = 0.5$ throughout preimplantation development till the 32-cell stage ($n = 332$ single cells from 42 embryos). In panels (**a**, **b**), the white dot indicates the median, the black bar indicates the first and third quartile, and the whiskers indicate the minima and maxima. **c** t-SNE map displaying two clusters of single cells at the 32-cell stage. These clusters were identified by $k$-means clustering on the 5mCpG strand bias for all paternal chromosomes (left). The right panel shows the strand bias variance within each cell superimposed on the t-SNE map. **d** The two clusters shown in panel (**c**) display dramatically different 5mCpG strand-bias distributions —one cluster (left) shows a unimodal distribution while the other cluster (right) shows a bimodal distribution implying loss of methylation maintenance. **e** Strand bias of chromosomes between anti-correlated cell pairs suggesting that these pairs are sister cells. **f** Bulk hairpin bisulfite sequencing reveals that the fraction of CpG dyads that are symmetrically methylated drops substantially from the 16- to 32-cell stage of development ($n = 2$ biologically independent bulk samples). Error bars represent the genome-wide standard deviation from the mean methylation maintenance.

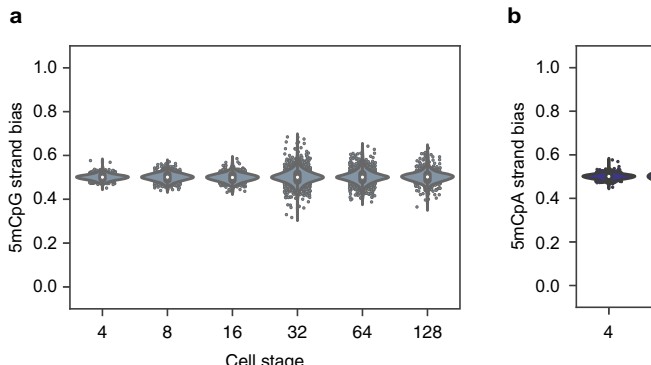

**Fig. 5 DNA demethylation dynamics in preimplantation human embryos. a** Violin plots showing 5mCpG strand bias from the 4- to 128-cell stage of human embryogenesis. In the absence of allele-specific information, the strand bias represents an average over both alleles. Similar to mouse embryos, human embryos initially show no 5mCpG strand bias followed by an increase at the 16-cell stage of embryogenesis. **b** Violin plots showing 5mCpA strand bias from the 4- to 128-cell stage of human embryogenesis. 5mCpA strand bias dynamics in human embryos is similar to that observed in mouse embryos in Fig. 4b. In these panels, the white dot indicates the median, the black bar indicates the first and third quartile, and the whiskers indicate the minima and maxima.

bias distributions of human preimplantation embryos appear to be similar to the trend observed in mouse embryos with a majority of cells till the 16-cell stage displaying 5mCpA strand bias (Fig. 5b and Supplementary Fig. 10b). Finally, upon closer inspection of 5mCpA strand bias per cell, we observed three sister pairs in day 3 embryos with a mirrored pattern of strand bias along the entire genome (Supplementary Fig. 10c).

In summary, we have developed a cost-effective and easy to implement strand-specific method that enables us to detect 5mC on a genome-wide scale in single cells. When applied to serum grown mES cells, we found substantial cell-to-cell variability in strand-specific 5mC landscapes, revealing the existence of chromosome-wide heterogeneity in the methylome of mES cells. Reanalysis of a previous single-cell bisulfite sequencing study further confirmed these results[25]. Furthermore, in addition to exploring strand-specific 5mC heterogeneity in single cells, scMspJI-seq also enables systematic investigation of the mechanisms regulating demethylation dynamics. In preimplantation mouse embryos, we surprisingly discovered two distinct phases of methylation dynamics—an initial phase till the 16-cell stage where methylation maintenance is active, followed by loss of maintenance in a fraction of cells within the early blastocyst at the 32-cell stage. These results further highlight the presence of strand-specific 5mC heterogeneity between individual cells during early mammalian development. In the future, we plan to explore how this genome-wide heterogeneity in the methylome regulates lineage commitment during development. Finally, despite the reduced resolution due to lack of allelic information, we found similar demethylation dynamics in preimplantation human embryos. Thus, scMspJI-seq presents a single-cell strand-specific technology that can potentially be used to probe the dynamics of methylation during development, cancer progression, aging, and in other biological systems.

## Methods

**Cell culture**. E14tg2a mES were obtained from American Type Culture Collection (ATCC CRL-182) and the hybrid 129/Sv:CAST/EiJ mES were obtained from Jop Kind's group (Hubrecht Institute). Both lines were tested for mycoplasma contamination. Cells were grown on 0.1% gelatin in ES cell culture media; DMEM (1×) high glucose + glutamax (Gibco), supplemented with 10% FCS (Greiner) 100 μM β-mercaptoethanol (Sigma), 100 μM Non-essential amino acids (Gibco), 50 μg/mL Pen/Strep (Gibco), and 1000 U/mL ESGRO mLIF (Millipore). Cells were split every 2 days and media changed every day. Cells were harvested before FACS by washing 3 times with 1× PBS with calcium and magnesium and incubated with 0.05% Trypsin (Life Technologies). Cell were resuspended in ES culture media and cell clumps were removed by passing the cells through a BD Falcon 5 mL polystyrene tube with a filter top.

**CRISPR-Cas9 *Dnmt1* knockout**. Six gRNA sequences targeting three exons of mouse *Dnmt1* were used as described previously[42]. Phosphorylated BbsI compatible restriction overhangs were added to gRNA top and bottom oligos and resuspended at 100 μM in nuclease-free water. Annealing of the oligos was performed in 1× ligation buffer (NEB) using the following program: 97 °C for 5 min, ramp down by 1 °C per 1 min to 20 °C. The pX330 CRISPR-Cas9-GFP gRNA plasmid was a kind gift from Eva van Rooij and mixed with 0.1 μM gRNA oligo. The reaction was simultaneously digested with BbsI (NEB) and ligated with T4 DNA ligase (NEB) overnight at 16 °C. Ligation reactions were transformed into DH5α competent cells and subsequently sequenced using Sanger dideoxy sequencing to confirm the correct insert. All six pX300-gRNA plasmids were pooled and 1 μg was transfected into 2 million E14tg2a cells using Lipofectamine (Life Technologies). A separate pX300 empty vector was also transfected into

E14tg2a to serve as a negative control. Two days later, single GFP-positive cells were sorted into 384-well plates (BioRad) and subjected to scMspJI-seq.

**Preimplantation mouse embryo isolation.** CAST/EiJ × C57BL/6 hybrid mouse embryos were obtained from four 3-month-old superovulated B6 mothers (injected with pregnant mare serum gonadotropin (PMSG) and human chorionic gonado-tropin (HCG) 22 h later), isolated using hyaluronic acid (Sigma), and incubated in M16 medium at 37 °C and 5% $CO_2$. The mice were housed at temperatures of 20–24 °C, humidity of 45–65%, and a light/dark cycle of 14/10 h. Individual cells were isolated using Tyrode's solution (Sigma) and trypsin (Life Technologies), and manually deposited into 384-well plates containing lysis buffer and Vapor-Lock. Plates were subsequently centrifuged at 1000 rpm for 1 min to ensure that cells reach the aqueous phase and then subjected to scMspJI-seq. All animal experiments were approved by the Royal Netherlands Academy of Arts and Sciences and were performed according to the animal experimentation guidelines of the KNAW.

**Preimplantation human embryo isolation.** Supernumerary cryopreserved human embryos were obtained for research from patients undergoing in vitro fertilization (IVF) using standard clinical protocols, at the Department for Reproductive Medicine, Ghent University Hospital. Cleavage stage embryos, cryopreserved on day 2 or 3 of development, were warmed using EmbryoThaw™ media (Fertipro, Belgium), as outlined by the manufacturer. Blastocyst stage embryos, vitrified on day 5 or 6 of development, were warmed using the Vitrification Thaw kit (Irvine Scientific, Netherlands), as described[43]. Embryos were transferred to either Cook Cleavage or Cook Blastocyst Medium (COOK, Ireland) depending on their developmental stage, and cultured in 20 µL medium droplets under mineral oil (Irvine Scientific, Netherlands) at 37 °C, 6% $CO_2$, and 5% $O_2$. When required, embryos were briefly treated with Acidic Tyrode's Solution (Sigma-Aldrich, Bel-gium) for removal of the zona pellucida. All embryos were washed and subse-quently dissociated by gentle mechanical dissociation in TrypLE Express Enzyme (Life Technologies, Belgium) using glass capillaries. Single blastomeres were washed and manually deposited into 384-well plates containing lysis buffer and Vapor-Lock. Plates were subsequently centrifuged at 1000 rpm for 1 min and stored at −80 °C until further processing. This study was approved by the Ghent University Institutional Review Board (EC2015/1114) and the Belgian Federal Commission for medical and scientific research on embryos in vitro (ADV_060_UZGent). All embryos were donated following patients' written informed consent.

**scMspJI-seq.** Prior to FACS or manual isolation of single cells, 384-well plates (BioRad) are prepared as follows: 4 µL of Vapor-Lock (Qiagen) is manually added to each well using a multichannel pipette followed by 2 µL of lysis buffer (0.2 µL of 25 µg/µL Qiagen Protease, 0.2 µL of 10× NEB Buffer 4 and 1.6 µL of nuclease-free water) using the Nanodrop II liquid-handling robot (BioNex Solutions). All downstream dispensing steps are performed using the liquid-handling robot. After spinning down the 384-well plates, single cells are deposited into each well of the plate and incubated at 50 °C for 15 h, 75 °C for 20 min, and 80 °C for 5 min. 5hmC sites in the genome are then glucosylated to block downstream recognition by MspJI by dispensing 0.5 µL of the following reaction mixture: 0.1 µL of T4-BGT (NEB), 0.1 µL of UDP-Glucose (NEB), 0.05 µL of 10× NEB Buffer 4, and 0.25 µL of nuclease-free water. After incubation at 37 °C for 16 h, 0.5 µL the following reaction mixture is added: 0.1 µL of 25 µg/µL Qiagen Protease, 0.05 µL of 10× NEB Buffer 4, and 0.35 µL of nuclease-free water. The plate is then incubated at 50 °C for 5 h, 75 °C for 20 min, and 80 °C for 5 min. Thereafter, gDNA is digested by the restriction enzyme MspJI by the addition of 0.5 µL of the following reaction mixture: 0.02 µL of MspJI (NEB), 0.12 µL of 30× enzyme activator solution (NEB), 0.05 µL of 10× NEB Buffer 4, and 0.31 µL of nuclease-free water. The digestion is performed at 37 °C for 5 h followed by heat inactivation of MspJI at 65 °C for 20 min. Next, 0.2 µL of cell-specific double-stranded adapters are added to indi-vidual wells and these adapters are ligated to the fragmented gDNA molecules by adding 0.8 µL of the following reaction mixture: 0.07 µL of T4 DNA ligase (NEB), 0.1 µL of T4 DNA ligase buffer (NEB), 0.3 µL of 10 mM ATP (NEB), and 0.33 µL of nuclease-free water. The ligation is performed at 16 °C for 16 h. Next, wells con-taining unique cell-specific adapters are pooled using a multichannel pipette and incubated with 0.8× Agencourt Ampure (Beckman Coulter) beads for 30 min, washed twice with 80% ethanol and resuspended in 6.4 µL of nuclease-free water. Thereafter, in vitro transcription and Illumina library preparation is performed as described previously in the scAba-seq protocol[22].

**scMspJI-seq adapters.** The double-stranded scMspJI-seq adapters are designed to contain a T7 promoter, 5′ Illumina adapter, 3 bp UMI, 8 bp cell-specific barcode, and a random 4-nucleotide 5′ overhang. The general design of the top and bottom strand is shown below:

Top oligo:
5′–CGATTGAGGCCGGTAATACGACTCACTATAGGGGTTCAGAGTTCT ACAGTCCGACGATCNNN[8 bp cell-barcode]–3′

Bottom oligo:
5′–NNNN[8 bp cell-barcode]NNNGATCGTCGGACTGTAGAACTCTGAA CCCCTATAGTGAGTCGTATTACCGGCCTCAATCG–3′

The sequence of the 8 bp cell-specific barcode is provided in Supplementary Table 1. The protocol for phosphorylating the bottom strand and for annealing the top and bottom strands to generate the double-stranded adapters is described previously in the scAba-seq protocol[22].

**scMspJI-seq analysis pipeline.** scMspJI-seq libraries were sequenced on an Illumina NextSeq 500 platform. Reads containing the correct cell-specific barcode were mapped to the mouse (mm10) or human (hg19) genome using the Burrows-Wheeler Aligner (BWA) and filtered for uniquely mapping reads to the genome. Custom scripts written in Perl were then used to demultiplex the data, identify 5mC position, strand information, and remove PCR duplicates. Custom code for analyzing scMspJI-seq data and the accompanying documentation is provided with this work (Supplementary Software).

**Strand-specific scNMT-seq analysis pipeline.** Bisulfite sequencing data from published scNMT libraries (GSE109262)[25] were processed as described pre-viously[44]. The first nine bases of the raw reads were trimmed using Trim Galore (v0.5.0) and mapped using Bismark (v20) to the mouse genome (mm10) with the 129/CAST background. SNPs specific to 129/CAST mouse genome were prepared using SNPsplit (v0.3.2) and a list of known variant call files from the Mouse Genomes Project (http://www.sanger.ac.uk/resources/mouse/genomes/). After mapping with Bismark, duplicate sequences were removed and CpG methylation calls were extracted with strand-specific information. Further data analysis and visualization of the methylation calls used custom scripts that will be made available upon request.

**Hairpin bisulfite sequencing.** Hairpin bisulfite sequencing was performed on bulk mouse embryos samples (2- to 64-cell stage mouse embryos). The embryos were treated with protease (1 µL of 25 µg/µL Qiagen Protease, 1 µL of 10× NEB Buffer 4, and 8 µL of nuclease-free water). Then, 0.5 ng of genomic DNA was digested with 20 µL of MspI master mix (1 µL of MspI (NEB), 2 µL 10× NEB CutSmart Buffer in a total volume of 20 µL) and incubated at 37 °C for 1 h. After digestion, the frag-mented genomic DNA was ligated with 1 µL of 10 µM phosphorylated hairpin oligo mix (1 µL of NEB T4 ligase, 1 µL of 10× NEB T4 Ligase buffer, 2 µL of 10 mM ATP, and 5 µL of nuclease-free water) and incubated overnight at 16 °C. The hairpin oligo was prepared as follows: The oligo (G/iMe-dC/iMe-dC/G/iMe-dC/ iMe-dC/GG/iMe-dC/GG/iMe-dC/AAG/iBiodT/GAAG/iMe-dC/iMe-dC/G/iMe-dC/iMe-dC/G/iMe-dC/G) was resuspended in 100 µM of Low-TE. The hairpin oligo was then phosphorylated (1 µL of 100 µM hairpin oligo, 3 µL of 10× T4 Ligase Buffer, 1 µL T4 PNK, and 5 µL of nuclease-free water) and incubated at 37 °C for an hour. Subsequently, the phosphorylated oligo was heated at 94 °C and placed in ice water to generate the loop. For purification of the ligation mixture, Dynabeads™ M-280 Streptavidin beads were used following the recommended manufacturer's protocol with the following changes: the bead-ligation mixture was incubated for 1 h at RT on a rotator and a cold 10 mM Tris-HCl wash step was included. Subsequently, we performed bisulfite sequencing on the sample using the protocol described previously[45]. After sequencing the libraries on a Miseq 300 bp or NextSeq 500 75 bp pair-end run, we used HBS-tools and custom Perl scripts to analyze the methylated CpG dyads[46].

**Reporting summary.** Further information on research design is available in the Nature Research Reporting Summary linked to this article.

## Data availability
All raw sequencing data has been uploaded to the Gene Expressio Omnibus under accession number "GSE139984". All other relevant data supporting the key findings of this study are available within the article and its Supplementary Information files or from the corresponding author upon reasonable request. A reporting summary for this Article is available as a Supplementary Information file.

## Code availability
Custom code for analyzing scMspJI-seq data and the accompanying documentation is provided with this work (Supplementary Software).

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

## Acknowledgements

We would like to thank members of the van Oudenaarden and Dey groups for constructive feedback. We thank patients of the Department for Reproductive Medicine, Ghent University Hospital, for donating their cryopreserved embryos for this study. We acknowledge support for the computational work from the Center for Scientific Computing at the California NanoSystems Institute (CNSI) and Materials Research Laboratory (MRL) at UCSB: an NSF MRSEC (DMR-1720256) and NSF CNS-1725797. We thank Ferring Pharmaceuticals (Aalst, Belgium) for an unrestricted educational grant. This work was supported by a Fonds Wetenschappelijk Onderzoek – Vlaanderen (FWO, Research Foundation – Flanders; G051516N) grant to B.H., the De Snoo-van't Hoogerhuijs Stichting to S.C.d.S.L., an European Research Council Advanced grant (ERC-AdG 742225-IntScOmics) and a Nederlandse Organisatie voor Wetenschappelijk Onderzoek (NWO) TOP award (NWO-CW 714.016.001) to A.v.O., and an UC Cancer Research Coordinating Committee (CTN-19-585462) grant and an NIH R01HD099517 grant to S.S.D. This work is part of the Oncode Institute which is partly financed by the Dutch Cancer Society.

## Author contributions

M.S., D.M., S.S.D., and A.v.O. designed the study. S.S.D. and D.M. developed the method. S.S.D., M.S., D.M., and A.C. performed experiments. J.-C.B. and D.M. isolated preimplantation mouse embryos. M.P., B.H., and S.C.d.S.L. obtained ethical permit and isolated preimplantation human embryos. M.S., A.C., D.M., J.-C.B., S.S.D., and A.v.O. analyzed the data. S.S.D., M.S., D.M., A.C., and A.v.O. wrote the manuscript.

## Competing interests

The authors declare no competing interests.
