## [Peer Review File · Nature Communications]

Reviewers' comments:

Reviewer #1 (Remarks to the Author):

Synopsis

Here, the authors describe a new single-cell technique for the strand-specific detection of 5mC and use it, first on ESCs as proof of principle. More importantly, they then apply it to early mouse embryos to revisit the proposed mechanism of maintenance/dilution of gametic 5mC marks from the 2-cell to 32-cell stages. The results are interesting and well presented. They offer a new approach to study epigenetic heterogeneity and their use of F1 material confirms that they can follow each of the four parental strands in the samples analyzed. Because sister cells exhibit reciprocal patterns upon loss of maintenance, or with maintenance but accompanied with mitotic recombination events, the technique can also trace pairs of sister cells, originating from a single mitosis. The key findings are also supported by hairpin bisulfite sequencing and extended to human embryos, which show a similar demethylation dynamics.

The manuscript is generally well written and present the results in a concise fashion. The interpretations and major conclusions are well supported by the data presented. Revisions are required to address the following points.

Comments

1. On line 66, the authors state "passive demethylation through the lack of Dnmt1 activity during replication" in the context of the passive demethylation of the maternal genome in preimplantation embryos. This statement is not correct. Although its activity might be restricted, DNMT1, and its cofactor UHRF1, are both present in small amount in the nucleus (Maenohara PLoS Genetics ,2017) and active during those stages, as it is known to be required for the maintenance of DNAm marks at imprinted genes and at several repetitive elements (Hirasawa, G&D, 2008; Smith, Nature, 2012).
2. I do not find Figure 1a really informative, since the authors are trying to distinguish between passive and active demethylation mechanisms. Although the bottom graph on the right illustrates the expectations for replication-coupled passive demethylation by lack of maintenance, the figure does not show what would be expected for an active, replication-independent mechanism. I would suggest adding a third graph on the right, directly linked to the left graph by an L-shaped arrow and showing, for instance, loss of 5mC at the middle CpG (or gain of 5hmC), while the flanking methylated CGs (5mC) are maintained on both strands.
3. For the data presented in Supplementary Figure 1a, only ~50% of methylated sites detected appear to be in the context of CpGs. Are the levels of non-CG methylation observed here similar to those detected by WGBS in ESCs? Is it known whether MspJI has any sequence preference for those two Ns in mCNR?
4. Although they authors claim that their technique can only detect 5mC, they do not actually show or calculate the efficiency of the glucosylation step that is required to block 5hmC sites. The authors need to provide some quantitative assessment of this step to clearly document the efficiency of this enzymatic step on genomic DNA. Otherwise the percentage of fall-positive 5mC calls is unknown. I understand that incomplete glucosylation would not affect their main insights on DNAm maintenance in preimplantation, since 5hmC is not maintained by DNMT1 anyway. However, since this is the first description of this technique, researchers will need a precise evaluation of this critical specificity-determining step.
5. How many unique 5mCG sites are detected per cell using this technique? They had provided this information for their related 5hmC detecting technique, but I could not find it here.
6. For their ESCs data, the authors should show the genomic distribution of these 5mC sites per class of genomic regions (CGI, promoter, exon, intro, intergenic, repetitive elements, etc) and

compare this with WGBS data. Since the technique is based on the generation of a cut 16 nucleotides 3' of a 5mC, it should bias against regions with closely spaced CpGs. Are CGIs or other CG-rich regions underrepresented in the data obtained? Although most CGI are unmethylated in ESCs, that is not the case for several of them in oocytes, and presumably in early embryos.

7. Are some imprinted DMRs detected in early embryos? These should be maintained on a single paternal allele. They should also be maintained throughout preimplantation development, so should represent exceptions to the dilution observed at the 32-cell stage. If data is available for 5mC within some of these regions, they should be included in the manuscript.

8. For the data generated by hairpin bisulfite sequencing (Fig. 4f), is it possible to represent the strand bias from the scMspJI-seq data as a single metric on the same graph? It looks like the former dataset suggest less maintenance from 2- to 16-cells. Is it because maternal and paternal alleles are not distinguished? Were the hairpin experiments performed on F1 embryos and if so, could the data be presented separately for each parental genome for reads including SNPs?

9. Line 224 - The first sentence of the "Discussion" states that the technique can quantify 5mC. What is the quantification aspect of the technique? Since only 5mC, and not C residues, are detected, how can the level of 5mC be quantified? To this reviewer, the technique appears mostly qualitative, especially for strand-specific information. Perhaps "to detect 5mC" would be more appropriate a term.

Minor comments

10. All protein names, such as DNMT1, should be in capital letters, no italics.

11. Figure 2d,e - Inconsistent numbering of cell # 562/526 in figure panels, legend, and text (line 119).
12. Figure 5a legend - "...human embryos initially show no 5mCpG strand [bias] followed by..."

Reviewer #2 (Remarks to the Author):

Sen et al present a novel epigenomics method, scMspJI-seq, which profiles genome-wide cytosine methylation in single-cells. The method is based on sensitivity of a restriction enzyme (MspJI) to cytosine modifications and is an adaptation of the group's previous method for profiling hydroxy-methyl-cytosine which used AbaSI. Single-cell bisulfite methods have been available for a number of years now but this method has a few key advantages which will make it more appropriate for some studies: firstly a simplified workflow which makes it more amenable to higher throughput, secondly a reduced sequencing burden (only 5mC sites are sequenced rather than the whole genome) means it is more cost-effective and finally the method is specific to 5mC rather than 5mC + 5hmC as with bisulfite. The authors first establish the method in mouse ESCs which is a cell type well studied using bisulfite based methods and thus makes comparisons easy. To demonstrate that they are able to detect strand-specific methylation, the authors compare WT ESCs to Dnmt1 KO ESCs, finding an increased strand-bias in cells lacking the maintenance methylation enzyme as expected. Next, the authors profile hybrid ESCs, using strain-specific SNPs to map their data to maternal/paternal chromosomes. This then allows them to assess strand-specific methylation at each individual chromosome. This led to the finding that a small population of ESCs have a strand bias in methylation, indicating a lack of maintenance methylation in these cells. The authors then re-analyse published scBS-seq data finding essentially the same result.

Next the authors perform scMspJI-seq on hybrid embryos to ask which stages undergo passive dilution of CpG methylation. They find a lack of CpG-methylation strand bias at early stages (in contrast CpA-methylation does display strand bias indicating that this difference is due to Dnmt1 activity which is specific to CpGs). At the 32-cell stages, strand bias increases suggesting reduced Dnmt1 activity and thus passive de-methylation. This result is then validated using hairpin-bisulfite

sequencing.

Overall the scMspJI-seq method is novel and expands on the available single-cell toolkit. The text is well written, the analysis rigorous and the biological findings nicely highlighted. In my view it is almost ready for publication, however, I have a few comments that I would like to see addressed first.

1. Regarding the quality of the data produced by the method. The authors state that 97.5% of sites overlap with published BS-seq - this is an important QC but this analysis is not actually presented in the paper. It should also be the case that a pseudobulk of their data gives quantitative information such that their data should correlate reasonably well with BS-seq data. Is this the case? Possibly coverage (in terms of number of cells x per cell coverage) is quite low which makes this analysis difficult but it should be presented. Otherwise we have no way of knowing the data produced is accurate.

2. In the mouse embryo analysis, the authors use CpA methylation as a control to show that strand bias is expected (for CpG methylation if Dnmt1 is not present). Can the authors comment on why CpA strand bias is not observed on the paternal chromosome?

3. The methods section is reasonably detailed with regards to the wetlab protocols. However, there is very little detail as to what was done to analyse the data. This needs to be included and the analysis code should be made available.

Reviewer #3 (Remarks to the Author):

In Sen et al the authors describe scMspJI-seq as a method for single-cell methylation analysis and apply the technique to assess strand bias as a measure of DNA methylation maintenance, using a DNMT1 knockout as a validation system. They then apply it to an interspecific mouse hybrid in vitro fertilization model as well as human preimplantation embryos. Overall the manuscript is very well written and polished. The analysis are sound and conclusions well-supported. A few aspects of phrasing could be improved – notably it is not a particularly high throughput method, despite the author's claims – it is still library construction on each individual cell in a well. If they had worked out a liquid handling solution to automate the process then perhaps the case could be made. Second, many existing single-cell bisulfite sequencing methods can capture strand information (e.g. <https://science.sciencemag.org/content/357/6351/600.abstract> – the first and second adaptors are incorporated in different ways, thus allowing strand discrimination). Also TAB-seq methods can distinguish between mC and hmC. Despite these details, which are minor and can be addressed by adjusting the text to be in-line with existing technologies – the technology appears to show some very interesting results and the application is very interesting. Below are more detailed comments:

One broad note – detecting strand bias in single-cell methylation sequencing using most methodologies is also possible. There is no aspect of this technology that makes it unique in that respect. The only advantage of this technique is that it is likely much cheaper than sc bisulfite methods; however it is likely at the cost of coverage – ie how many CpGs are profiled per cell. If this is comparable and the cost is substantially lower, then that is a selling point.

Line 78 – bisulfite-based methods can identify the difference between 5mC and 5hmC using TET-based assays where the readout is either 5mC+5hmC or just 5hmC alone. This is the well-known TAB-seq method. The authors should remove all reference to bisulfite methods being unable to distinguish between the two.

Line 84 – typo: "genome" -> "genomic"

The authors need to include some of the metrics for QC on the technology in the main text. E.g. how many detected methylated cytosines per cell? How does it compare to sc-mC methods? What

is the distribution in the genome? What is the distribution of covered CpGs per cell? What is the meta-gene methylation profile? What about meta-CpG-island profile? I am sure the authors have all of these numbers / profiles, so including them is important.

The strand-bias metric is simple and informative and conveys the information clearly.

The knockout of DNMT1 is an excellent way to validate the approach for characterizing DNA methylation dynamics.

The use of hairpin bisulfite sequencing is a solid way to confirm the observation of reduced methylation maintenance at the 16-32 cell stage.

For the human dataset – is it possible to quantify aneuploidy? This is a frequent phenomenon in primates as well as a number of other mammals (not mouse) and would be an interesting analysis to tie into the methylation analysis.

Response to Reviewer #1

We thank Reviewer #1 for her/his insightful and constructive suggestions, which has significantly improved the manuscript. Remarks of Reviewer #1 are denoted in *italics*.

Synopsis: Here, the authors describe a new single-cell technique for the strand-specific detection of 5mC and use it, first on ESCs as proof of principle. More importantly, they then apply it to early mouse embryos to revisit the proposed mechanism of maintenance/dilution of gametic 5mC marks from the 2-cell to 32-cell stages. The results are interesting and well presented. They offer a new approach to study epigenetic heterogeneity and their use of F1 material confirms that they can follow each of the four parental strands in the samples analyzed. Because sister cells exhibit reciprocal patterns upon loss of maintenance, or with maintenance but accompanied with mitotic recombination events, the technique can also trace pairs of sister cells, originating from a single mitosis. The key findings are also supported by hairpin bisulfite sequencing and extended to human embryos, which show a similar demethylation dynamics.

The manuscript is generally well written and present the results in a concise fashion. The interpretations and major conclusions are well supported by the data presented. Revisions are required to address the following points.

We thank Reviewer #1 for finding our results interesting and well supported by the data presented. We have addressed the comments of Reviewer #1 in detail below:

1. *On line 66, the authors state “passive demethylation through the lack of Dnmt1 activity during replication” in the context of the passive demethylation of the maternal genome in preimplantation embryos. This statement is not correct. Although its activity might be restricted, DNMT1, and its cofactor UHRF1, are both present in small amount in the nucleus (Maenohara PLoS Genetics ,2017) and active during those stages, as it is known to be required for the maintenance of DNAm marks at imprinted genes and at several repetitive elements (Hirasawa, G&D, 2008; Smith, Nature, 2012).*

We had discussed the existence of two isoforms of Dnmt1 during preimplantation development, including their nuclear localization status, later in the manuscript (lines 166-171). We noted that the isoform Dnmt1s is found at low levels within the nucleus of blastomeres that could be responsible for maintaining DNA methylation on the maternal genome during preimplantation development. We thank the reviewer for pointing out that this was not clear earlier in the manuscript. Based on the reviewers suggestion, we have now included the Maenohara *et al.* citation (Hirasawa *et al.* 2008 *Genes Dev.* and Smith *et al.* 2012 *Nature* were already cited in the initial submission), and edited sentences in the manuscript as follows (lines 64-73):

“While early immunofluorescence-based studies revealed that the paternal genome undergoes active demethylation through conversion to 5hmC in the zygote, the maternal genome was presumed to undergo passive demethylation through the lack of DNMT1 activity during replication. Advances in biochemistry, next-generation sequencing and mass spectroscopy based studies improved upon this coarse quantification of methylation dynamics to show that the orthogonal regulation of demethylation by active and passive mechanisms for the two parental genomes was not as distinct as suggested by these early studies. For example, it was later shown that while DNMT1 is mostly cytoplasmic during these early stages of development, low levels of a *Dnmt1* isoform, DNMT1s, together with UHRF1 is observed in the nucleus, raising the possibility that 5mC is maintained on the maternal genome.”

2. I do not find Figure 1a really informative, since the authors are trying to distinguish between passive and active demethylation mechanisms. Although the bottom graph on the right illustrates the expectations for replication-coupled passive demethylation by lack of maintenance, the figure does not show what would be expected for an active, replication-independent mechanism. I would suggest adding a third graph on the right, directly linked to the left graph by an L-shaped arrow and showing, for instance, loss of 5mC at the middle CpG (or gain of 5hmC), while the flanking methylated CGs (5mC) are maintained on both strands.

We thank the reviewer for pointing this out. We have now changed Figure 1a by adding another schematic panel showing that a combination of symmetric DNA methylation on both DNA strands of a chromosome coupled with the observation of loss of DNA methylation by bulk bisulfite sequencing would indicate active demethylation (Figure R1). Accordingly, we have also changed the legend for Figure 1a as follows (lines 540-543):

“Cells displaying symmetric levels of 5mCpG on both DNA strands of a chromosome coupled with a global temporal loss of 5mCpG indicates active demethylation whereas loss of methylation maintenance with asymmetric levels of 5mCpG between the two DNA strands indicates passive demethylation.”

Figure R1 | Schematic of DNA demethylation dynamics

3. For the data presented in Supplementary Figure 1a, only ~50% of methylated sites detected appear to be in the context of CpGs. Are the levels of non-CG methylation observed here similar to those detected by WGBS in ESCs? Is it known whether MspJI has any sequence preference for those two Ns in mCNR?

Using scMspJI-seq, we observe that an average of 44.0% (with a range of 31.0% to 59.9% in individual cells) of the methylated cytosines in E14 mESCs are in a non-CpG context. To address the reviewers question, we analyzed single-cell whole-genome bisulfite sequencing data (scWGBS) in E14 mESCs from Smallwood *et al*¹. We have updated Supplementary Figure 1 with an additional panel showing this new analysis (Figure R2 below and Supplementary Fig. 1b in the manuscript). Compared to scMspJI-seq, we find a similar distribution of methylated cytosines in a non-CpG context with CpA methylation being the most prevalent, as observed previously². In the scWGBS data for E14 cells, we find that 31.5% (with a range of 23.4% to 53.1% in individual cells) of the methylated cytosines are in a non-CpG context. These results suggest that scMspJI-seq and scWGBS detect a similar number of methylated cytosines in a non-CpG context with the differences in detection rate possibly arising due to cell culture differences or due to MspJI having

a slight sequence preference for non-CpG methylated sites over CpG methylated sites. Finally, MspJI has previously been shown to not display any substantial sequence preference at the two Ns within ^mCNNR³.

Figure R2 | Analysis of nucleotide composition downstream of the methylated site in bisulfite sequencing.

4. Although they authors claim that their technique can only detect 5mC, they do not actually show or calculate the efficiency of the glucosylation step that is required to block 5hmC sites. The authors need to provide some quantitative assessment of this step to clearly document the efficiency of this enzymatic step on genomic DNA. Otherwise the percentage of fall-positive 5mC calls is unknown. I understand that incomplete glucosylation would not affect their main insights on DNAm maintenance in preimplantation, since 5hmC is not maintained by DNMT1 anyway. However, since this is the first description of this technique, researchers will need a precise evaluation of this critical specificity-determining step.

In E14 mESCs, comparison of 5mC strand bias using scMspJI-seq to 5hmC strand bias using scAba-seq from our previous work shows that the 5mC strand bias displays a tight distribution centered around 0.5, whereas the 5hmC strand bias distribution is wide ranging from 0.1 to 0.9 (Figure R3a,b)⁴. These results qualitatively suggest that we are primarily detecting 5mC sites in the genome in scMspJI-seq, consistent with the maintenance activity of DNMT1 in E14 cells. In scMspJI-seq, we use T4 phage β -glucosyltransferase (T4- β GT) to convert 5hmC to glucosylated sites in the genome, blocking downstream detection of these sites by MspJI. Previous mass spectroscopy based quantification has shown that T4- β GT converts 5hmC to glucosylated 5hmC at efficiencies close to 100%⁵⁻⁷. Furthermore, only 5% of CpG sites in mESCs are hydroxymethylated; therefore, taken together this suggests that we are predominantly detecting 5mC in scMspJI-seq and that our results are not impacted by the efficiency of 5mC detection. Finally, to address the reviewers question, we combined data from scMspJI-seq and scAba-seq to quantify the false positive detection rate of 5hmC in scMspJI-seq. For different efficiencies of 5mC vs. 5hmC detection, we built a mathematical model where 5mC and 5hmC sites were drawn from a binomial distribution and distributed on the two DNA strands of a chromosome using the strand bias distributions from scMspJI-seq and scAba-seq. By comparing the variance of the experimental strand bias distribution to those obtained from the simulations, we estimated that the false-positive detection rate of 5hmC is around 1.1% (Figure R3c, Revised manuscript:

Supplementary Fig. 7). We have included the results of this simulation in the revised manuscript (lines 116-119):

“In addition, compared to bisulfite sequencing, an advantage of scMspJI-seq is that it can identify only 5mC in the genome by blocking detection of 5hmC sites using T4-βGT. By combining scMspJI-seq data with scAba-seq results, we were able to estimate the false-positive detection rate of 5hmC to be around 1.1% (Supplementary Fig. 7).”

Figure R3 | Quantifying the false positive detection rate of 5hmC in scMspJI-seq.

5. How many unique 5mCG sites are detected per cell using this technique? They had provided this information for their related 5hmC detecting technique, but I could not find it here.

We thank the reviewer for bringing this to our attention. For the E14 mESCs, the number of unique 5mC sites detected per cell ranged from 212,000 to 977,000, with a median of 484,000 5mC sites per cell. Further, we observed that the number of unique 5mC sites detected per cell increases monotonically as a function of the sequencing depth, suggesting that more unique sites could be detected per cell by sequencing the Illumina libraries deeper (Figure R4). We have included the panel shown below as Supplementary Figure 2 and the following text in the main manuscript (lines 101-103):

“We detected between 212,000 to 977,000 unique 5mC sites per cell, with a median of 484,000 5mC sites per cell (Supplementary Fig. 2)”.

Figure R4 | Number of unique 5mC sites detected per cell as a function of the sequencing depth in scMspJI-seq.

6. For their ESCs data, the authors should show the genomic distribution of these 5mC sites per class of genomic regions (CGI, promoter, exon, intro, intergenic, repetitive elements, etc) and compare this with WGBS data. Since the technique is based on the generation of a cut 16 nucleotides 3' of a 5mC, it should bias against regions with closely spaced CpGs. Are CGIs or other CG-rich regions underrepresented in the data obtained? Although most CGI are unmethylated in ESCs, that is not the case for several of them in oocytes, and presumably in early embryos.

As the reviewer suggested, we have now compared the genome-wide distribution of 5mC in scMspJI-seq and scWGBS over different genomic elements. Overall, we see that both methods display a similar distribution of 5mC over different genomic regions (Figure R5, Revised manuscript: Supplementary Fig. 4). However, we observe that scMspJI-seq is biased against the detection of 5mC sites within exons, whereas it detects more sites within introns compared to scWGBS. As the reviewer points out, this could possibly occur as scMspJI-seq is dependent on the digestion of the genome around methylated cytosines, which could bias our technique against detection of closely spaced 5mC sites. We explored this more carefully to show that scMspJI-seq shows a slight preference for detection of 5mC sites within genomic regions that have a lower density of CpG sites. In contrast, scWGBS shows a slight preference for detection of 5mC sites within genomic regions that have a higher density of CpG sites (Figure R6, Revised manuscript: Supplementary Fig. 5). Despite these differences between the two methods, we capture similar genome-wide landscapes of 5mC for a variety of genomic elements. For example, as expected, we observe hypomethylation at CpG islands (CGI) and transcription start sites (TSS) in scMspJI-seq, similar to that observed in published scWGBS (Figure R7a-d, Revised manuscript: Supplementary Fig. 6a-d). We also find that the meta-gene methylome profiles are similar for scMspJI-seq and scWGBS (Figure R7e,f, Revised manuscript: Supplementary Fig. 6e,f). Corresponding to these additional figure panels, we have added the following text to the main manuscript (lines 107-115):

“Furthermore, while we observed that the genome-wide distribution of 5mC over different genomic elements in scMspJI-seq was similar to that observed in bisulfite sequencing, we also found that scMspJI-seq shows a slight preference for detection of 5mC sites within genomic regions that have a lower density of CpG sites (Supplementary Fig. 4,5). This possibly occurs as our method is dependent on the digestion of the genome around methylated cytosines, reducing the likelihood of detecting closely spaced 5mC sites. However, both scMspJI-seq and bisulfite sequencing captured similar genome-wide landscapes of 5mC at a variety of genomic elements. For example, we observed similar gene body methylome profiles as well as the expected hypomethylation of CpG islands (CGI) and transcription start sites (TSS) using both methods (Supplementary Fig. 6).”

Figure R5 | Distribution of 5mC over different genomic elements in scMspJI-seq and bisulfite sequencing.

Figure R6 | Distribution of 5mCpG sites over genomic regions of varying CpG density in scMspJI-seq and scWGBS.

Figure R7 | Genome-wide DNA methylation landscapes over CpG islands, transcription start sites and gene bodies obtained from scMspJI-seq and scWGBS.

7. Are some imprinted DMRs detected in early embryos? These should be maintained on a single paternal allele. They should also be maintained throughout preimplantation development, so

should represent exceptions to the dilution observed at the 32-cell stage. If data is available for 5mC within some of these regions, they should be included in the manuscript.

Based on the reviewers suggestion, we analyzed 5mC levels at imprinted genes. However, we find that we typically detect fewer than 30 5mC marks/cell within these genes (Figure R8). Given this low resolution, we were unable to test if these regions represent exceptions to the global passive demethylation we observe from the 16- to 32-cell stage of development. While we believe this is outside the scope of the current work, it is an excellent suggestion that we plan to follow up on in the future. Therefore, we have not included this analysis and figure in the revised manuscript.

Figure R8 | Box plots of the number of 5mC marks detected per cell within imprinted genes during preimplantation mouse development.

8. For the data generated by hairpin bisulfite sequencing (Fig. 4f), is it possible to represent the strand bias from the scMspJI-seq data as a single metric on the same graph? It looks like the former dataset suggest less maintenance from 2- to 16-cells. Is it because maternal and paternal alleles are not distinguished? Were the hairpin experiments performed on F1 embryos and if so, could the data be presented separately for each parental genome for reads including SNPs?

Bulk hairpin bisulfite sequencing enables interrogation of the methylation status of individual CpG dyads whereas scMspJI-seq enables strand-specific quantification of 5mC in individual chromosomes of single cells. As the methylation status of individual CpG dyads in hairpin bisulfite sequencing cannot be used to estimate the strand-specific methylation levels of an individual chromosome, it is not possible to directly represent the strand bias from scMspJI-seq on Figure 4f. The bulk hairpin bisulfite sequencing experiments were not performed on hybrid embryos. However, we do not expect the slight reduction in the fraction of symmetrically methylated CpG dyads from the 2- to 16-cell stage to arise from our inability to distinguish the parental alleles. We have now clearly indicated in the manuscript that the experiments were performed on non-hybrid embryos (lines 220-222):

“Finally, to conclusively demonstrate that the absence of 5mCpG strand bias up to the 16-cell stage arises from DNMT1 mediated maintenance methylation, we performed bulk hairpin bisulfite sequencing on non-hybrid preimplantation mouse embryos.”

9. Line 224 - The first sentence of the “Discussion” states that the technique can quantify 5mC. What is the quantification aspect of the technique? Since only 5mC, and not C residues, are detected, how can the level of 5mC be quantified? To this reviewer, the technique appears mostly

qualitative, especially for strand-specific information. Perhaps “to detect 5mC” would be more appropriate a term.

scMspJI-seq is useful for measuring the relative levels of 5mC between two strands of a chromosome. We thank the reviewer for her/his suggestion and we have now changed the text in the main manuscript as follow (lines 244-245):

“In summary, we have developed a new high-throughput strand-specific method that enables us to detect 5mC on a genome-wide scale in single cells.”

Minor comments

10. *All protein names, such as DNMT1, should be in capital letters, no italics.*

We thank the reviewer for pointing out this inconsistency in nomenclature. We have corrected this in the entire manuscript.

11. *Figure 2d,e - Inconsistent numbering of cell # 562/526 in figure panels, legend, and text (line 119).*

We thank the reviewer for pointing out this typographical error. We have now corrected the legend of Figure 2a to ensure consistent numbering (lines 554-556).

12. *Figure 5a legend - “...human embryos initially show no 5mCpG strand [bias] followed by...”*

We have now corrected this typographical error (line 597).

Response to Reviewer #2

We thank Reviewer #2 for her/his valuable suggestions, which we have incorporated in the revised manuscript. Remarks of Reviewer #2 are denoted in *italics*.

Sen et al present a novel epigenomics method, scMspJI-seq, which profiles genome-wide cytosine methylation in single-cells. The method is based on sensitivity of a restriction enzyme (MspJI) to cytosine modifications and is an adaptation of the group's previous method for profiling hydroxy-methyl-cytosine which used AbaSI. Single-cell bisulfite methods have been available for a number of years now but this method has a few key advantages which will make it more appropriate for some studies: firstly a simplified workflow which makes it more amenable to higher throughput, secondly a reduced sequencing burden (only 5mC sites are sequenced rather than the whole genome) means it is more cost-effective and finally the method is specific to 5mC rather than 5mC + 5hmC as with bisulfite. The authors first establish the method in mouse ESCs which is a cell type well studied using bisulfite based methods and thus makes comparisons easy. To demonstrate that they are able to detect strand-specific methylation, the authors compare WT ESCs to Dnmt1 KO ESCs, finding an increased strand-bias in cells lacking the maintenance methylation enzyme as expected. Next, the authors profile hybrid ESCs, using strain-specific SNPs to map their data to maternal/paternal chromosomes. This then allows them to assess strand-specific methylation at each individual chromosome. This led to the finding that a small population of ESCs have a strand bias in methylation, indicating a lack of maintenance methylation in these cells. The authors then re-analyse published scBS-seq data finding essentially the same result.

Next the authors perform scMspJI-seq on hybrid embryos to ask which stages undergo passive dilution of CpG methylation. They find a lack of CpG-methylation strand bias at early stages (in contrast CpA-methylation does display strand bias indicating that this difference is due to Dnmt1 activity which is specific to CpGs). At the 32-cell stages, strand bias increases suggesting reduced Dnmt1 activity and thus passive de-methylation. This result is then validated using hairpin-bisulfite sequencing.

Overall the scMspJI-seq method is novel and expands on the available single-cell toolkit. The text is well written, the analysis rigorous and the biological findings nicely highlighted. In my view it is almost ready for publication, however, I have a few comments that I would like to see addressed first.

We thank the reviewer for highlighting the advantages of scMspJI-seq and for finding the manuscript almost ready for publication. We have addressed the comments of Reviewer #2 in detail below:

1. Regarding the quality of the data produced by the method. The authors state that 97.5% of sites overlap with published BS-seq - this is an important QC but this analysis is not actually presented in the paper. It should also be the case that a pseudobulk of their data gives quantitative information such that their data should correlate reasonably well with BS-seq data. Is this the case? Possibly coverage (in terms of number of cells x per cell coverage) is quite low which makes this analysis difficult but it should be presented. Otherwise we have no way of knowing the data produced is accurate.

We thank the reviewer for these suggestions. We have now included a Venn diagram that shows the overlap in the methylated sites detected by bulk bisulfite sequencing and scMspJI-seq (Figure R9a, Revised manuscript: Supplementary Fig. 3a). Further, we find that the averaged single-cell MspJI-seq data correlates well with published bulk bisulfite sequencing (Pearson $r = 0.84$ for genomic bin size of 1 MB) (Figure R9b, Revised manuscript: Supplementary Fig. 3b).

Corresponding to these additional figure panels, we have added the following text to the main manuscript (lines 103-106):

“Further, we found that 97.2% of the 5mCpG sites detected by scMspJI-seq in single cells overlapped with methylated sites observed in bulk bisulfite sequencing of E14 gDNA (Supplementary Fig. 3a). Similarly, we found that averaged single-cell data from scMspJI-seq correlates well with the bulk bisulfite methylome (Pearson $r = 0.84$) (Supplementary Fig. 3b)”

In addition, we find that the genome-wide distribution of 5mC at different genomic elements is similar for scMspJI-seq and whole-genome bisulfite sequencing (Figure R5, Revised manuscript: Supplementary Fig. 4). We also show that the methylome landscapes at different genomic elements, such as CpG islands, transcription start sites and gene bodies, are similar for scMspJI-seq and scWGBS (Figure R7, Revised manuscript: Supplementary Fig. 6). For details, please see response to Reviewer #1, comment 6.

Figure R9 | Comparison of scMspJI-seq to bulk bisulfite sequencing

2. In the mouse embryo analysis, the authors use CpA methylation as a control to show that strand bias is expected (for CpG methylation if *Dnmt1* is not present). Can the authors comment on why CpA strand bias is not observed on the paternal chromosome?

The 5mCpA strand bias is dependent on both the initial amount of CpA methylation present on the original parental DNA strands, and the rate of *de novo* CpA methylation on the new DNA strands. The original maternal chromosomes have been shown to contain over 5-fold more non-CpG methylation than the original paternal chromosomes⁸. Therefore, we hypothesize that with the same *de novo* methylation rates for both the maternal and paternal genomes, 5mCpA strand bias is observed at the 2-cell stage on the maternal genome but not on the paternal genome because the newly synthesized DNA strands potentially achieve CpA methylation levels that are much lower than that present on the original maternal DNA strands, whereas it is comparable to that present on the original paternal DNA strands.

3. The methods section is reasonably detailed with regards to the wetlab protocols. However, there is very little detail as to what was done to analyse the data. This needs to be included and the analysis code should be made available.

The custom code to analyze scMspJI-seq data, and accompanying documentation to run the code and interpret the output files was provided during the original submission. However, this was not stated in the main manuscript. We thank the reviewer for pointing this out. We have now clearly directed readers to this in the Methods and Code Availability section (lines 369-370 and 408-409):

“Custom code for analyzing scMspJI-seq data and the accompanying documentation is provided with this work”

Response to Reviewer #3

We thank Reviewer #3 for her/his suggestions, which we have included in the revised manuscript. Remarks of Reviewer #3 are denoted in *italics*.

In Sen et al the authors describe scMspJI-seq as a method for single-cell methylation analysis and apply the technique to assess strand bias as a measure of DNA methylation maintenance, using a DNMT1 knockout as a validation system. They then apply it to an interspecific mouse hybrid in vitro fertilization model as well as human preimplantation embryos. Overall the manuscript is very well written and polished. The analysis are sound and conclusions well-supported. A few aspects of phrasing could be improved – notably it is not a particularly high throughput method, despite the author’s claims – it is still library construction on each individual cell in a well. If they had worked out a liquid handling solution to automate the process then perhaps the case could be made. Second, many existing single-cell bisulfite sequencing methods can capture strand information (e.g. <https://science.sciencemag.org/content/357/6351/600.abstract> – the first and second adaptors are incorporated in different ways, thus allowing strand discrimination). Also TAB-seq methods can distinguish between mC and hmC. Despite these details, which are minor and can be addressed by adjusting the text to be in-line with existing technologies – the technology appears to show some very interesting results and the application is very interesting.

We thank the reviewer for finding our results well supported by the data and for finding the manuscript well written. We would like to clarify the following points that the reviewer has raised above:

(1) scMspJI-seq is performed using a robotic platform as described in the Methods section (lines 324-325). Single cells within individual wells of a 384-well plate are processed using the liquid-handling robot. Therefore, such a platform allows us to process thousands of single cells per day in a high-throughput format.

(2) Illumina library preparation is not performed on individual cells. As described in the Methods section (lines 341-345), ligation of double-stranded adapters to fragmented gDNA barcodes individual cells. Ligated gDNA molecules are then pooled and processed in a single tube to prepare Illumina libraries. Thus, the combination of cellular barcoding and the use of a liquid-handling robot enables high-throughput processing in scMspJI-seq.

(3) We agree with the reviewer that certain bisulfite sequencing approaches can capture strand-specific information. In recognition of this, during our original submission we reanalyzed published single-cell bisulfite sequencing data from hybrid mESCs to show that a small population of cells show significant 5mC strand bias, similar to that observed in our data using scMspJI-seq (Figure 3 and lines 150-157).

(4) While we agree with the reviewer that TAB-seq can distinguish between 5mC and 5hmC, this method cannot be directly extrapolated to a single-cell resolution. This is because TAB-seq measurements for detecting only 5mC sites in the genome require splitting up the gDNA sample, which is clearly not possible in single cells.

In addition to these notes, we have addressed the comments of Reviewer #3 in detail below:

Below are more detailed comments:

1. One broad note – detecting strand bias in single-cell methylation sequencing using most methodologies is also possible. There is no aspect of this technology that makes it unique in that respect. The only advantage of this technique is that it is likely much cheaper than sc bisulfite

methods; however it is likely at the cost of coverage – ie how many CpGs are profiled per cell. If this is comparable and the cost is substantially lower, then that is a selling point.

For details on the number of the unique 5mC sites detected per cell, please see our response to Reviewer #1, comment 5. While the coverage of scWGBS is slightly higher than scMspJI-seq¹, we find that the number of unique 5mC sites detected per cell in scMspJI-seq increases monotonically with the sequencing depth in the current libraries, suggesting that more unique 5mC sites can be detected by sequencing the libraries deeper. Comparable coverage of the methylomes, together with the relative ease and high-throughput nature of scMspJI-seq, makes this new method well suited to detect strand-specific methylation in single cells.

2. Line 78 – bisulfite-based methods can identify the difference between 5mC and 5hmC using TET-based assays where the readout is either 5mC+5hmC or just 5hmC alone. This is the well-known TAB-seq method. The authors should remove all reference to bisulfite methods being unable to distinguish between the two.

Starting from bulk gDNA, as the reviewer mentions, TAB-seq can either be used to quantify the combination of 5mC and 5hmC or just 5hmC alone. Therefore, in order to identify 5mC sites, the bulk gDNA has to be split in two parts, where one half is used to detect 5mC+5hmC whereas the other half is used to detect 5hmC alone. Subtracting the results from these two datasets enables quantification of just 5mC sites in the genome. However, gDNA from single cells cannot be subjected to two independent measurements and therefore, unlike scMspJI-seq, TAB-seq cannot be used to detect only 5mC sites in single cells. However, we agree with the reviewer that our phrase “bisulfite-based methods” in the original submission might be confusing, and therefore we have changed the text in the revised manuscript to “bisulfite sequencing” as follows (lines 81-83):

“Bisulfite sequencing cannot distinguish between 5mC and 5hmC, and importantly when applied to bulk cellular populations, this approach cannot be used to quantify the relative levels of 5mC between two strands of a single chromosome”.

3. Line 84 – typo: “genome” -> “genomic”

We thank the reviewer for identifying this typographical error. This has been corrected in the revised manuscript (line 87).

4. The authors need to include some of the metrics for QC on the technology in the main text. E.g. how many detected methylated cytosines per cell? How does it compare to sc-mC methods? What is the distribution in the genome? What is the distribution of covered CpGs per cell? What is the meta-gene methylation profile? What about meta-CpG-island profile? I am sure the authors have all of these numbers / profiles, so including them is important.

For details on the number of 5mC sites detected per cell in scMspJI-seq, please see response to Reviewer #1, comment 5 and Supplementary Figure 2.

For details on the genome-wide distribution of 5mC sites in scMspJI-seq and how it compares to whole-genome bisulfite sequencing, please see response to Reviewer #2, comment 1 and Reviewer #1, comment 6, and Supplementary Figures 3-5.

For details on 5mC meta-profiles for different genomic elements, such as gene bodies, CpG islands and transcription start sites, in scMspJI-seq and its comparison to whole-genome bisulfite sequencing, please see response to Reviewer #1, comment 6 and Supplementary Figure 6.

5. The strand-bias metric is simple and informative and conveys the information clearly.

We thank the reviewer for finding the strand bias metric informative for conveying the results in this manuscript.

6. *The knockout of DNMT1 is an excellent way to validate the approach for characterizing DNA methylation dynamics.*

We are glad that the reviewer found the *Dnmt1* knockout experiments useful in validating scMspJI-seq.

7. *The use of hairpin bisulfite sequencing is a solid way to confirm the observation of reduced methylation maintenance at the 16-32 cell stage.*

We thank the reviewer for finding the hairpin bisulfite sequencing results as conclusive demonstration of the reduction in maintenance methylation from the 16- to 32-cell stage of preimplantation development.

8. *For the human dataset – is it possible to quantify aneuploidy? This is a frequent phenomenon in primates as well as a number of other mammals (not mouse) and would be an interesting analysis to tie into the methylation analysis.*

To address the reviewers question, we analyzed single cells from human preimplantation blastocysts by quantifying the number of 5mC sites detected in 10 MB bins along the human genome. We then used the circular binary segmentation algorithm to call copy numbers variations in single cells⁹. While we find that the majority of cells within human blastocysts are diploid, we also detect a small fraction of cells that display aneuploidy as the reviewer suggested (Figure R10). This is an interesting result; however, we believe this observation is outside the scope of this manuscript and we have therefore not included this panel in the revised manuscript.

Figure R10 | Using scMspJI-seq to call genomic copy number variations in single cells from human blastocysts.

References:

1. Smallwood, S. A. *et al.* Single-cell genome-wide bisulfite sequencing for assessing epigenetic heterogeneity. *Nat. Methods* **11**, 817–820 (2014).
2. Ramsahoye, B. H. *et al.* Non-CpG methylation is prevalent in embryonic stem cells and may be mediated by DNA methyltransferase 3a. *Proc. Natl. Acad. Sci. U.S.A.* **97**, 5237–5242 (2000).
3. Cohen-Karni, D. *et al.* The MspJI family of modification-dependent restriction endonucleases for epigenetic studies. *Proc. Natl. Acad. Sci. U.S.A.* **108**, 11040–11045 (2011).
4. Mooijman, D., Dey, S. S., Boisset, J. C., Crosetto, N. & van Oudenaarden, A. Single-cell 5hmC sequencing reveals chromosome-wide cell-to-cell variability and enables lineage reconstruction. *Nat. Biotechnol.* **34**, 852–856 (2016).
5. Robertson, A. B. *et al.* A novel method for the efficient and selective identification of 5-hydroxymethylcytosine in genomic DNA. *Nucleic Acids Res.* **39**, e55 (2011).
6. Vaisvila, R. *et al.* EM-seq: Detection of DNA Methylation at Single Base Resolution from Picograms of DNA. *bioRxiv* doi: <https://doi.org/10.1101/2019.12.20.884692>.
7. Sun, Z. *et al.* A sensitive approach to map genome-wide 5-hydroxymethylcytosine and 5-formylcytosine at single-base resolution. *Mol. Cell* **57**, 750–761 (2015).
8. Wang, L. *et al.* Programming and Inheritance of Parental DNA Methylomes in Mammals. *Cell* **157**, 979–991 (2014).
9. Venkatraman, E. S. & Olshen, A. B. A faster circular binary segmentation algorithm for the analysis of array CGH data. *Bioinformatics* **23**, 657–663 (2007).

REVIEWERS' COMMENTS

Reviewer #1 (Remarks to the Author):

I would like to thank the authors for addressing all my comments and performing additional analyses and figures in the process. Nice job!

Reviewer #2 (Remarks to the Author):

The authors have addressed my comments in full and I would be happy to recommend this work for publication.

Reviewer #3 (Remarks to the Author):

“(1) scMspJI-seq is performed using a robotic platform as described in the Methods section (lines 324-325). Single cells within individual wells of a 384-well plate are processed using the liquid handling robot. Therefore, such a platform allows us to process thousands of single cells per day in a high-throughput format.”

I missed that – including the use of a liquid handling platform in the main will highlight this.

“(2) Illumina library preparation is not performed on individual cells. As described in the Methods section (lines 341-345), ligation of double-stranded adapters to fragmented gDNA barcodes individual cells. Ligated gDNA molecules are then pooled and processed in a single tube to prepare Illumina libraries. Thus, the combination of cellular barcoding and the use of a liquid handling robot enables high-throughput processing in scMspJI-seq.”

The ligation of barcoded adaptors is a key library preparation step – granted some aspects are performed after pooling, there are indeed library preparation steps that are processing a single cell in a single well as opposed to droplet based cell barcoding etc...

“(3) We agree with the reviewer that certain bisulfite sequencing approaches can capture strand specific information. In recognition of this, during our original submission we reanalyzed published single-cell bisulfite sequencing data from hybrid mESCs to show that a small population of cells show significant 5mC strand bias, similar to that observed in our data using scMspJI-seq (Figure 3 and lines 150-157).”

This was a good addition and I believe strengthens the manuscript by having that concordance.

“(4) While we agree with the reviewer that TAB-seq can distinguish between 5mC and 5hmC, this method cannot be directly extrapolated to a single-cell resolution. This is because TAB-seq measurements for detecting only 5mC sites in the genome require splitting up the gDNA sample, which is clearly not possible in single cells.”

True, but the alternative is instead of splitting gDNA, cells are split to go through each “arm” of the protocol, which is the same strategy used here where cells are compared to infer those differences.

“Bisulfite sequencing cannot distinguish between 5mC and 5hmC, and importantly when applied to bulk cellular populations, this approach cannot be used to quantify the relative levels of 5mC between two strands of a single chromosome .”

The authors state this, but the same would be true for bulk MspJI-seq, and also agree that strand bias can be detected in single-cell bisulfite libraries. Based on that, I feel this statement is misleading.

Line 151: "While this method is low throughput, single-cell bisulfite sequencing can potentially also be used to infer strand-specific 5mC" – the authors state that it is low throughput, but it is the same throughput as scMspJI-seq. Cell barcoding is performed on single cells in single wells then subsequent steps are performed in a pool. Plus they use liquid handling robots. There is no difference in throughput with technologies like the ones used by the Ecker Lab for instance compared to scMspJI-seq. There is no inherent throughput advantage of scMspJI-seq over existing single-cell bisulfite methods and all references that state the contrary should be removed.

I am not sure why the authors focus on throughput, when there is no real advantage there. However, there are actual advantages of their technology that they have every right to focus on. This includes a far more amenable method when compared to difficult and unwieldy bisulfite methods. I also expect that not having bisulfite reads means better alignment and fewer reads that are thrown out. There are real advantages, throughput is just not one of them, and the paper stands on its own without that. The only notes on throughput can be that scMspJI-seq is set up to be performed on a liquid handling robot which enables high throughput – not that it is higher throughput than bisulfite methods.

Back to the TAB-seq argument – on a single cell one can detect either 5mC+5hmC OR 5hmC alone; I agree that one cannot split the gNDA of a single cell and compare. However, scMspJI-seq can be used to assess either 5mC alone or 5mC+5hmC (if the T4-BGT step is omitted). Both platforms assess different sets but *neither* can assess both in the same cell where they can be detected independently. I.e. Knowing in a single cell which sites are 5mC and which are 5hmC. I do not think the authors need to say things are not possible using other methods, when they actually are in a comparable way. They should instead highlight that they can obtain these different sets, similar to bisulfite methods, but their technology is far more cost effective, which is an important point.

Again – I really think this is an excellent technology that stands on its own – but there are many misleading claims that it does things better than bisulfite when that is simply not true. It does do things better than bisulfite – namely the ease of the method and cost effectiveness – which is a big deal and should be emphasized.

I stand by my previous comments that the application that the authors present is excellent. The analysis is well done, clearly presented, and has very interesting findings.

I appreciate the copy number analysis – the fraction of cells that are aneuploid match what would be expect – though I would not call it a small fraction. However, I agree that it is outside the scope of this manuscript, though I do hope the authors explore this further – I would be very interested to see methylation differences in these aneuploid blastocysts.

Remarks of the reviewers are denoted in *italics*.

Response to Reviewer #1

I would like to thank the authors for addressing all my comments and performing additional analyses and figures in the process. Nice job!

We would like to thank the reviewer for their constructive feedback that has significantly improved our manuscript.

Response to Reviewer #2

The authors have addressed my comments in full and I would be happy to recommend this work for publication.

We would thank the reviewer for recommending our work for publication.

Response to Reviewer #3

1. *“(1) scMspJI-seq is performed using a robotic platform as described in the Methods section (lines 324-325). Single cells within individual wells of a 384-well plate are processed using the liquid handling robot. Therefore, such a platform allows us to process thousands of single cells per day in a high-throughput format.”*

I missed that – including the use of a liquid handling platform in the main will highlight this.

We have now included a statement in the main manuscript to clarify that scMspJI-seq is performed using a liquid handling platform (lines 87-88).

“All downstream steps are subsequently performed using a liquid-handling platform (Nanodrop II, BioNex Solutions).”

2. *“(2) Illumina library preparation is not performed on individual cells. As described in the Methods section (lines 341-345), ligation of double-stranded adapters to fragmented gDNA barcodes individual cells. Ligated gDNA molecules are then pooled and processed in a single tube to prepare Illumina libraries. Thus, the combination of cellular barcoding and the use of a liquid handling robot enables high-throughput processing in scMspJI-seq.”*

The ligation of barcoded adaptors is a key library preparation step – granted some aspects are performed after pooling, there are indeed library preparation steps that are processing a single cell in a single well as opposed to droplet based cell barcoding etc...

We agree with the reviewer that the initial steps (up to ligation of cell-specific double-stranded adapters to fragmented genomic DNA) are performed on single cells that is followed by pooling, *in vitro* transcription and downstream Illumina library preparation. However, these initial steps of the protocol are executed on the liquid-handling platform, and therefore it is straightforward to process hundreds to thousands of single cells per day. We agree with the reviewer though that droplet-based barcoding methods or single-cell combinatorial indexing methods are generally higher throughput methods. Therefore, as the term “high-throughput” is not quantitative, we have removed this term throughout the manuscript and we do not describe scMspJI-seq as a high-

throughput method. Instead, we have added a sentence to clearly indicate the throughput of scMspJI-seq (lines 98-100):

“The ligated molecules are then amplified by *in vitro* transcription and used to prepare Illumina libraries as described previously, enabling the processing of hundreds to thousands of single cells per day (Fig. 1b)”.

3. *“(3) We agree with the reviewer that certain bisulfite sequencing approaches can capture strand specific information. In recognition of this, during our original submission we reanalyzed published single-cell bisulfite sequencing data from hybrid mESCs to show that a small population of cells show significant 5mC strand bias, similar to that observed in our data using scMspJI-seq (Figure 3 and lines 150-157).”*

This was a good addition and I believe strengthens the manuscript by having that concordance.

We are glad that the reviewer thinks that the concordance between the single-cell bisulfite sequencing and scMspJI-seq data strengthens the manuscript.

4. *“(4) While we agree with the reviewer that TAB-seq can distinguish between 5mC and 5hmC, this method cannot be directly extrapolated to a single-cell resolution. This is because TAB-seq measurements for detecting only 5mC sites in the genome require splitting up the gDNA sample, which is clearly not possible in single cells.”*

True, but the alternative is instead of splitting gDNA, cells are split to go through each “arm” of the protocol, which is the same strategy used here where cells are compared to infer those differences.

When measurements are made in individual cells, bisulfite sequencing provides a combined readout of 5mC and 5hmC, while scMspJI-seq provides a readout of 5mC only. If TAB-seq were implemented at a single-cell resolution, it would provide a readout of 5hmC alone in individual cells. However, most importantly, we believe this discussion on TAB-seq does not impact the manuscript as this work focuses entirely on 5mC and does not mention/discuss TAB-seq or quantification of 5hmC in single cells.

5. *“Bisulfite sequencing cannot distinguish between 5mC and 5hmC, and importantly when applied to bulk cellular populations, this approach cannot be used to quantify the relative levels of 5mC between two strands of a single chromosome .”*

The authors state this, but the same would be true for bulk MspJI-seq, and also agree that strand bias can be detected in single-cell bisulfite libraries. Based on that, I feel this statement is misleading.

We agree with the reviewer that when starting with bulk populations, none of the methods can quantify the relative levels of 5mC between two strands of a single chromosome. We thank the reviewer for pointing this out and therefore, we have now removed the following sentence from the manuscript:

“Bisulfite sequencing cannot distinguish between 5mC and 5hmC, and importantly when applied to bulk cellular populations, this approach cannot be used to quantify the relative levels of 5mC between two strands of a single chromosome.”

6. Line 151: “While this method is low throughput, single-cell bisulfite sequencing can potentially also be used to infer strand-specific 5mC” – the authors state that it is low throughput, but it is the same throughput as scMspJI-seq. Cell barcoding is performed on single cells in single wells then subsequent steps are performed in a pool. Plus they use liquid handling robots. There is no difference in throughput with technologies like the ones used by the Ecker Lab for instance compared to scMspJI-seq. There is no inherent throughput advantage of scMspJI-seq over existing single-cell bisulfite methods and all references that state the contrary should be removed.

I am not sure why the authors focus on throughput, when there is no real advantage there. However, there are actual advantages of their technology that they have every right to focus on. This includes a far more amenable method when compared to difficult and unwieldy bisulfite methods. I also expect that not having bisulfite reads means better alignment and fewer reads that are thrown out. There are real advantages, throughput is just not one of them, and the paper stands on its own without that. The only notes on throughput can be that scMspJI-seq is set up to be performed on a liquid handling robot which enables high throughput – not that it is higher throughput than bisulfite methods.

In our experience, when we have performed single-cell bisulfite sequencing in our lab (following the protocol of Smallwood *et al.* Nat. Methods 2014), we have found the method to be more labor intensive. However, following the reviewer’s suggestion, we have removed any direct throughput comparison of single-cell bisulfite sequencing and scMspJI-seq. Accordingly, we do not refer to single-cell bisulfite sequencing as a low throughput method, and we have changed the main manuscript as follows (lines 155-157):

“To validate this cell-to-cell heterogeneity in 5mC strand bias, we reanalyzed data from a recent study that quantified 5mC in single cells using bisulfite sequencing, a method that can potentially also be used to infer strand-specific 5mC.”

Further, following the reviewer’s suggestion, we have highlighted that scMspJI-seq is performed on a liquid handling platform enabling a throughput of hundreds to thousands of single cells per day, without referring to scMspJI-seq as having higher throughput than single-cell bisulfite sequencing (please see response to Reviewer #3, comments 1 and 2). Finally, as recommended by the reviewer, we have now included a sentence stating the ease of implementation of scMspJI-seq (please see response to Reviewer #3, comment 8).

7. Back to the TAB-seq argument – on a single cell one can detect either 5mC+5hmC OR 5hmC alone; I agree that one cannot split the gNDA of a single cell and compare. However, scMspJI-seq can be used to assess either 5mC alone or 5mC+5hmC (if the T4-BGT step is omitted). Both platforms assess different sets but *neither* can assess both in the same cell where they can be detected independently. Ie Knowing in a single cell which sites are 5mC and which are 5hmC. I do not think the authors need to say things are not possible using other methods, when they actually are in a comparable way. They should instead highlight that they can obtain these different sets, similar to bisulfite methods, but their technology is far more cost effective, which is an important point.

We agree with the reviewer that when applied to single cells, bisulfite sequencing can be used to detect the combination of 5mC and 5hmC, and TAB-seq could be used to detect 5hmC alone. Further, scMspJI-seq can be used to detect 5mC alone or a combination of 5mC and 5hmC (if the T4-βGT step is omitted) in single cells. We agree completely with the reviewer that these methods detect different sets of DNA modifications and that these methods are complimentary

with distinct strengths and advantages. Accordingly, we have changed the following sentences in the main manuscript to indicate the different sets of DNA modifications that are detected by single-cell bisulfite sequencing and scMspJI-seq (lines 90-91 and 118-120):

We have changed the sentence “This modification blocks downstream detection of 5hmC and therefore, unlike bisulfite-based methods enables detection of only 5mC.” to “This modification blocks downstream detection of 5hmC and therefore, enables detection of only 5mC in scMspJI-seq”.

Similarly, we have changed the sentence “In addition, compared to bisulfite sequencing, an advantage of scMspJI-seq is that it can identify only 5mC in the genome by blocking detection of 5hmC sites using T4-βGT.” to “In addition, compared to single-cell bisulfite sequencing that detects a combination of 5mC and 5hmC sites, a distinct feature of scMspJI-seq is that it can identify only 5mC in the genome by blocking detection of 5hmC sites using T4-βGT.”

8. Again – I really think this is an excellent technology that stands on its own – but there are many misleading claims that it does things better than bisulfite when that is simply not true. It does do things better than bisulfite – namely the ease of the method and cost effectiveness – which is a big deal and should be emphasized.

We thank the reviewer for finding our technology excellent. As suggested by the reviewer, we have now modified our discussion on the throughput of scMspJI-seq (please see response to Reviewer #3, comments 1 and 2) as well as altered our comparison between bisulfite sequencing and scMspJI-seq (please see response to Reviewer #3, comments 4, 5, 6 and 7). Further, as recommended by the reviewer, we have now edited the following sentence to highlight the cost effectiveness and ease of scMspJI (lines 251-252):

“In summary, we have developed a new cost effective and easy to implement strand-specific method that enables us to detect 5mC on a genome-wide scale in single cells.”

9. I stand by my previous comments that the application that the authors present is excellent. The analysis is well done, clearly presented, and has very interesting findings.

We thank the reviewer for finding our analysis rigorous and results interesting.

10. I appreciate the copy number analysis – the fraction of cells that are aneuploid match what would be expect – though I would not call it a small fraction. However, I agree that it is outside the scope of this manuscript, though I do hope the authors explore this further – I would be very interested to see methylation differences in these aneuploid blastocysts.

We would like to thank the reviewer again for pointing us in this direction and we plan to follow up on this work in the future.